# MamBEV: Enabling State Space Models to Learn Birds-Eye-View Representations

Hongyu Ke[1]*    Jack Morris[1]*    Kentaro Oguchi[2]    Xiaofei Cao[2]    Yongkang Liu[2]
Haoxin Wang[1]    Yi Ding[1]
[1]Georgia State University
[2]InfoTech Labs, Toyota Motor North America R&D
{hke3, jmorris116, haoxinwang, yiding}@gsu.edu
{kentaro.oguchi, xiaofei.cao, yongkang.liu}@toyota.com

## Abstract

3D visual perception tasks, such as 3D detection from multi-camera images, are essential components of autonomous driving and assistance systems. However, designing computationally efficient methods remains a significant challenge. In this paper, we propose a Mamba-based framework called MamBEV, which learns unified Bird's Eye View (BEV) representations using linear spatio-temporal SSM-based attention. This approach supports multiple 3D perception tasks with significantly improved computational and memory efficiency. Furthermore, we introduce SSM based cross-attention, analogous to standard cross attention, where BEV query representations can interact with relevant image features. Extensive experiments demonstrate MamBEV's promising performance across diverse visual perception metrics, highlighting its advantages in input scaling efficiency compared to existing benchmark models. The code is available at https://github.com/amai-gsu/MamBEV.

## 1 Introduction

Automatically constructing a bird's-eye-view (BEV) of an object's surrounding environment is beneficial for tasks such as autonomous driving and driver assistance systems (Wang et al., 2023a). These methods typically integrate the signals received by multi-view cameras and transforms them into a top-down view of the surrounding environment. Furthermore, as these systems operate in an mobile edge environment, it is important to consider the computational costs in conjunction with construction accuracy (Ke et al., 2024).

Examples of deployed BEV systems can be seen in Tesla cars (Tesla, 2021). These detailed constructions can be used for downstream perceptual, prediction, and planning tasks (Casas et al., 2021; Hu et al., 2023). There are two predominant methods for building BEV models: push and pull. Push methods project 2D image features into 3D space using pixel-wise depth predictions, transforming flat images into spatially-aware 3D representations. Pull methods, on the other hand, sample points from a 3D prior and project them onto the 2D image plane, allowing the model to extract 3D information without explicit depth predictions (Li et al., 2022a; Wang et al., 2023b; Chen et al., 2022; 2023). However, many of these methods rely on the use of transformers' costly attention mechanism to learn accurate representations. Thus, we are motivated to examine more efficient methods that can replace transformer-based architectures.

Recently, Gu et al. (2021) and Gu & Dao (2023) showed that state space models (SSMs) can replace the quadratic computational complexity of transformers' attention through a linear approximation. These models have been shown to work comparably to transformers at scale on large language models (Dao & Gu, 2024) such as Codestral Mamba. There has also been evidence that SSM's attention can replace transformer attention to learn effective visual representations (Zhu et al., 2024; Liu et al., 2024; Patro & Agneeswaran, 2024). While Mamba, like self-attention, is effective at capturing intra-sequence dependencies, it struggles in BEV scenarios involving multiple input modalities, where dynamic information exchange is crucial.

---

*Equal Contribution.

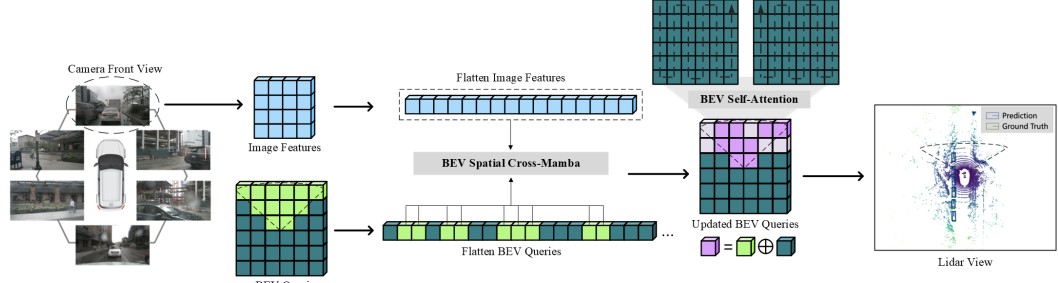

Figure 1: We propose MamBEV, a novel paradigm that leverages both SSM based Cross-Attention and Self-Attention mechanisms to generate BEV features from multi-camera inputs.

Motivated by these findings, we examine how SSMs can be used to generate a BEV representation. This is important because it is costly to capture temporal and spatial relationships in multiview videos. We examine how the linear attention inside SSMs can be used to address these issues. However, incorporating SSMs into 3D representation learning tasks is not well understood. Furthermore, it is unclear how to fuse distinct visual representations as this is a crucial step in learning BEV representations.

Our paper attempts to address these issues through the following contributions:

- We propose an SSM-based architecture, MamBEV, that can exceed the performance of existing Transformer-based architectures.

- We propose a novel approach, Spatial Cross Mamba, analogous to standard cross-attention, where a set mapping mechanism enables the association and fusion of two different modalities. In our case, BEV query representations are matched with corresponding image features to facilitate direct integration of information from both modalities.

- A thorough set of ablation studies is provided to showcase model scaling and other properties. We open-source our code [1] and provide a strong baseline and evaluation framework for future experimentation.

## 2 BACKGROUND

### 2.1 LEARNING BEV REPRESENTATIONS

BEV representations are widely used in autonomous driving as they provide a dense unified representation of a scene which can be used for a variety of tasks. One approach, based on LSS (Philion & Fidler, 2020), involves projecting 2D images into a 3D space by leveraging the camera intrinsics, such as Li et al. (2023b), Li et al. (2023a), Han et al. (2024). For example, VideoBEV (Han et al., 2024) adopts an LSS-based 3D BEV detection approach, incorporating a customized recurrent-style temporal fusion and a temporal embedding block to leverage long-term information more effectively. However, this method heavily depends on accurate depth estimation to ensure the precision of feature projections. On the other hand, BEVFormer Li et al. (2022a) introduces a direct approach by pulling features from 2D image space into a BEV space, collapsing all height information into a single BEV query. It adopts deformable attention Zhu et al. (2020), a sparse sampling method, as the core of the BEV encoder which maps multiview image features to BEV features. BEVFormerV2 Yang et al. (2023) added an additional head and image augmentations to improve supervision for the model's backbone. It also uses a CNN based temporal encoder in place of the recurrent deformable temporal attention used in BEVFormer.

Deformable attention offers the advantage of reduced memory consumption, but it also has notable limitations. For example, deformable attention provides only sparse and limited supervision for the backbone, as it does not engage with all image features Yang et al. (2023). In terms of feature refinement, deformable attention relies heavily on the backbone to produce strong feature representations. For instance, in the Deformable Attention Transformer (DAT) Xia et al. (2022), multiheaded full

---

[1]https://github.com/amai-gsu/MamBEV

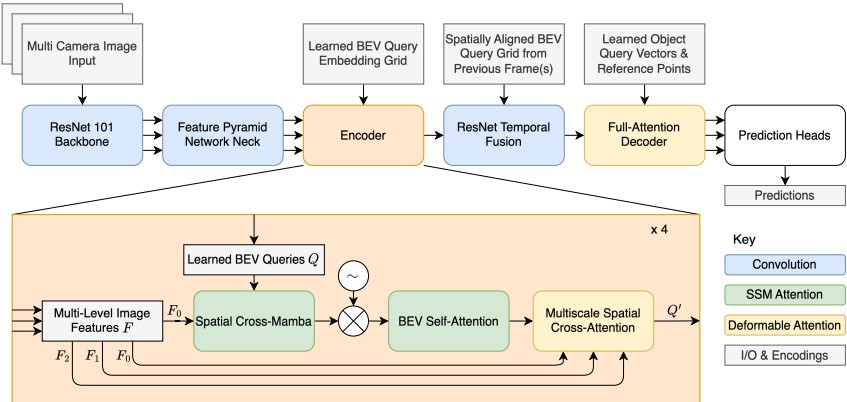

Figure 2: The overall pipeline of of our architecture (MamBEV-Small). We present a novel method for incorporating SSMs into a BEV construction algorithm. Features are extracted from six egocentric multiview camera images over multiple frames. A ResNet backbone is used to extract camera features which are passed to SSM based encoder blocks. We found that it was necessary to use full attention during the decoding process, however this has limited impact on the computational complexity as the encoded feature sequence is relatively short.

attention remains necessary to refine features effectively, underscoring the limitations of deformable attention in refining spatial representations. From a hardware perspective, deformable attention is suboptimal for modern GPU architectures due to its reliance on random memory access during the sampling process, which negatively impacts model throughput Zhu et al. (2020).

## 2.2 LINEAR ATTENTION WITH STATE SPACE MODELS

State space models (SSMs), particularly structured SSMs, present an effective linear complexity alternative to the prevalent transformer architecture for sequence modeling. At their core, SSMs operate on data from each channel independently which updates a latent state $h$ that evolves across the input sequence. The evolution of this latent state is governed by a set of learnable parameters, denoted as $A, B, C$. Mamba Gu & Dao (2023) introduced selective SSMs (S6) which enables the model to dynamically change the values of the $A, B, C$ parameter matrices based on the input, effectively acting as a filter or gating mechanism. This mechanism better allows SSMs to model input sequences which have elements with varying degrees of information density, such as text, as it can selectively ignore tokens which are information sparse and selectively remember those which are information dense. Mamba also included significant hardware optimizations which allows for parallel computation of state updates using an associative scan.

Mamba-2 (Dao & Gu, 2024) leverages a novel understanding of the connection between SSMs and attention mechanisms, termed state space duality (SSD), to overcome some key weaknesses of its predecessor, Mamba-1. SSD demonstrates that SSM computations can be expressed through a dual form involving structured matrix multiplication. This allows Mamba-2 to leverage highly optimized matrix multiplication units on GPUs, resulting in a significant speedup compared to Mamba-1's scan-based implementation.

While Mamba-2 excels in autoregressive tasks, its underlying SSM framework inherently operates in a causal manner, limiting its applicability to non-causal scenarios. Hydra (Hwang et al., 2024) addresses this limitation by leveraging quasiseparable matrix mixers which generalize the semiseparable matrix mixer found in Mamba-2 to encompass both lower and upper triangular components. This change enables bidirectional information flow with a minimal increase in computation and parameters. For simplicity, we use Mamba when discussing the Hydra block in the following sections of the paper, as Hydra uses the same block structure and SSM formulation as Mamba-2.

**SSM Inner Function.** State Space Models can be considered as systems that map a signal $x(t) \in \mathbb{R}$ into $y(t) \in \mathbb{R}$ through $h(t) \in \mathbb{R}^N$, which can be formulated as

$$h'(t) = Ah(t) + Bx(t), \quad y(t) = Ch(t) + Dx(t), \tag{1}$$

where $A \in \mathbb{R}^{N \times N}$ is the evolution parameter, $B \in \mathbb{R}^{N \times 1}$, $C \in \mathbb{R}^{1 \times N}$ and $D \in \mathbb{R}$ are the projection parameters.

**Discretized Inner Function.** In machine learning applications, most inputs are not continuous signals so to adapt these systems to discrete input sequences, the system itself must be discretized. The discretized version can be formulated as

$$h_t = \bar{A}h_{t-1} + \bar{B}x_t, \tag{2}$$
$$y_t = Ch_t + Dx_t, \tag{3}$$

where a timescale parameter $\Delta$ is used to transform the continuous parameters $A$ and $B$ to discrete parameters $\bar{A}$ and $\bar{B}$:

$$\Delta = \text{softplus}(dt + dt_{bias}), \tag{4}$$
$$\bar{A} = e^{\Delta A}, \tag{5}$$
$$\bar{B} = (\Delta A)^{-1}(e^{\Delta A} - I)\Delta B. \tag{6}$$

This operation explicitly ties the hidden state update parameters $A$ and $B$ together. In practice, this operation is not necessary since the model can learn the discretized system directly as noted in (Gu & Dao, 2023). However, the inclusion of a discretization function provides a useful mechanism for interpreting how a given input interacts with the hidden state as well as methods which enable the direct control over whether the hidden state is updated by a given token or class of tokens. Ali et al. (2024) notes that each channel having a distinct discretization parameter is equivalent to having multiple heads in attention where each head in this case corresponds to its own channel. In Mamba-2, the authors found that tying multiple channels together by sharing a discretization factor between them reduced computational complexity while retaining a similar level of expressivity. When grouping multiple channels into a head, the $A$ matrix can be shared and the number of $dt$ which are computed as a function of the inputs are also reduced. Additionally, they found that reducing $A$ from a diagonal matrix in $\mathbb{R}^{N \times N}$ to a scalar reduced computational cost with minimal impact on expressivity. The SSM component in diagrams refer to the operations in equations 2 and 3.

## 3 METHODS

The BEV construction problem is a supervised learning algorithm which maps camera images $\mathcal{I}_t = \{m_1, ..., m_k\}$ of $k$ camera views at a particular time step $t$ onto a 2D planar Birds-Eye-View of objects. A given BEV scene that contains $l$ objects, has bounding boxes $\{b_1, ...b_l\} \in \mathcal{B}$, classes $\{c_1, ..., c_l\} \in \mathcal{C}$, as well as trajectory information $\mathcal{T}$. The approach is to first learn an encoding transformation $Q = f_{enc}(\mathcal{I}_t)$, for all $t$, to obtain a latent BEV query representation $Q \in \mathbb{R}^{H \times W \times D}$ where $H, W$ represents the spatial shape of the BEV grid and $D$ is the latent dimension. A decoding transformation is then learned to predict object information $\mathcal{B}, \mathcal{C}, \mathcal{T} = f_{dec}(Q)$ by sampling the BEV grid.

**BEV Method Overview.** A visualization of our overall method for learning a BEV construction is provided in Figure 2. Many of the existing components follow the methodologies of Li et al. (2022a), Li et al. (2023b), and Liu et al. (2023). The encoding function $f_{enc}$ is parameterized by the ResNet101 backbone, feature pyramid network (FPN), BEV encoder, and temporal fusion modules. The pipeline has three main stages as follows. First, image feature maps of different scales $F_0, ..., F_i$ are produces as intermediate backbone outputs. Then their channels are reduced to a uniform size $D$ through horizontal convolutions in the FPN (Lin et al., 2017). Second, feature representations and interactions are modeled using a novel SSM-based pipeline (discussed below). Lastly, the decoding function $f_{dec}$ operates on this representation to make predictions. As in Zhu et al. (2020) Li et al. (2022a) Yang et al. (2023) methods, the decoder uses alternating layers of grouped object query self-attention and object query to BEV feature deformable cross-attention. Decoder heads at each layer follow the masked decoder heads from Li et al. (2022b) which predict a bounding box, its properties, and the object class. We do not use a map segmentation head, and only use the detection head which is optimized using 3D hungarian set matching with smooth $L_1$ loss for the bounding boxes and focal loss for class prediction.

We next discuss the novel encoder seen in Figure 2.

## 3.1 SPATIAL CROSS MAMBA

The central challenge in using Mamba to as a BEV is how to perform cross attention in an efficient manner. There are simple cross attention adaptations for SSMs which are inefficient or not well suited for the problem. For example, by expanding the state size $N$ to equal the number of image features $T$ Mamba-2 is able to store all information about previous tokens at the cost of returning to complexity of transformers Dao & Gu (2024). To perform this naive form of mamba cross attention, first compute the final hidden state $h_T \in \mathbb{R}^{T \times \alpha D}$ over the image features then find $y_Q$ using

$$y_i = C_i h_T \qquad (7)$$

where $C_i \in \mathbb{R}^{1 \times T}$ is a function of a corresponding query $q_i \in Q$. Computing $h_T$ takes $TNP = T^2 P$ FLOPs where $P$ is the latent dimension per head and $TN = T^2$ memory which in this case is equivalent to the FLOPs and memory used to compute self-attention in a Transformer, $T^2 N$ and $T^2$. To compute $y_i$ requires $T|Q|\alpha D$ FLOPs and $T|Q|$ memory giving a total computational and memory complexity similar to Transformer cross attention. To reduce the computational cost from the naive implementation, we propose task specific adaptations as shown in Figure 3 which allow us to reduce the size of the hidden dimension $N << T$.

### 3.1.1 REDUCING STATE SIZE

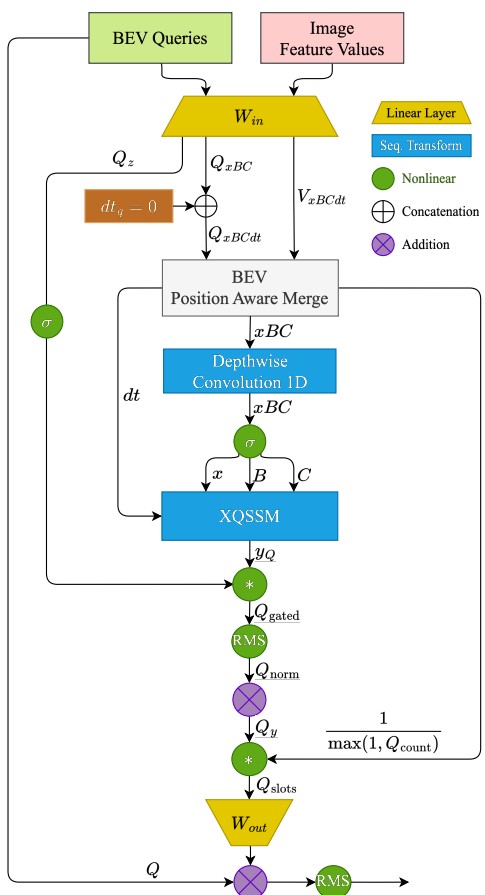

Figure 3: Proposed Spatial Cross Mamba using XQSSM. Our novel method to fuse two distinct spatial representations: 1) BEV queries which is a top-down representation, and 2) image features which come from an egocentric view.

Simply reducing $N$ may still give acceptable results if the image is information sparse, however this cannot be guaranteed. The resulting $h_T$ would likely lose information about image features near the start of the sequence. This results from the definition of the decay parameter $\exp(\Delta A) \in [0, 1], A < 0, \Delta \geq 0$, which guarantees that the impact of $x_0$ on $y_T$ decreases as $T$ increases except when $\Delta_i = 0, i = 1, \ldots, T$. The solution we propose is to compute $y_i$ using $C_i h_k$ where image feature vector $k$ is likely to be relevant to $q_i$. Since multiple image regions may be relevant to a single BEV query, we use $Z$ copies of $C_i$ to attend to $Z$ locations on the image feature map, then

$$y_i = \sum_{k=0}^{Z} C_i h_k. \qquad (8)$$

To select the $Z$ feature map locations for a query $q_i$ we lift the 2D BEV location (x, y) into a 3D pillar (x,y,z) and project $Z$ evenly spaced pillar points onto each image Lang et al. (2018). The resulting locations called reference points $R \in \mathbb{R}^{|Q| \times Z \times 2}$ correspond to where an object at a BEV location would appear in the image based on the ego vehicle's camera calibration and assuming there are no obstructions. The number of locations which fall inside the image bounds, $R_{\text{hit}} = \{r_{ij} \mid r_{ij} \in [0, 1]^2\} \subseteq R$ is significantly smaller than $QZ$. We refer to the size of $R_{\text{hit}}$ as $M = f(|Q|, Z, \theta_{\text{FOV}}) \approx \frac{ZQ}{\# \text{ of cameras}}$ as most $q_i$ are only visible in a single view.

### 3.1.2 BEV POSITION AWARE MERGE

$Q$, and $V$ are flattened to 1D sequences so that they can be processed by Mamba. $V$ is flattened according to a traversal order $\mathcal{T} : \mathbb{R}^{H \times W \times \alpha D} \mapsto \mathbb{R}^{HW \times \alpha D}$. Then $Q_{\text{hit}}$ are merged with the flattened

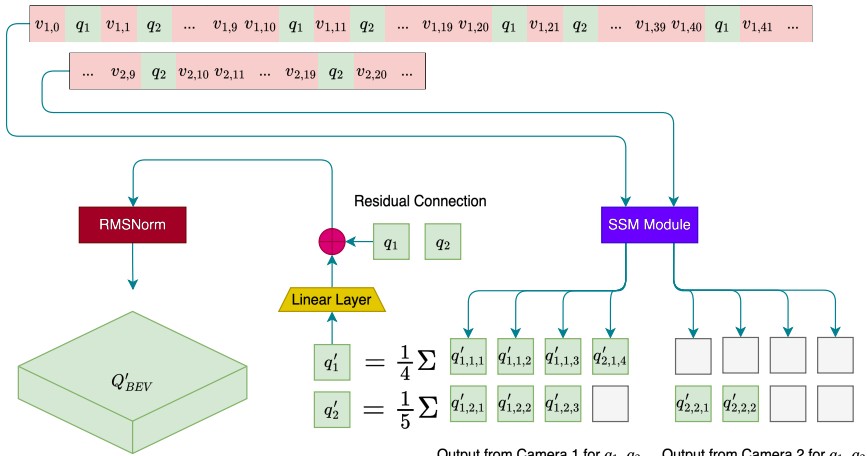

Figure 4: Spatial Cross Mamba Pre- and Post-processing. Illustration of the processing performed on the input and output of the SSM to merge sampled information from multiple query copies in the input sequence into the BEV query grid. Image features (denoted by $v_{i,j}$) and their corresponding query vectors ($q_k$) are first interleaved to enable causal attention. Processed outputs of the SSM are normalized and fused into an updated query matrix $Q'_{BEV}$.

sequence of image features $F$ through the 1D reference points $R_{1D}$ as follows:

$$R_{1D} := h\lfloor wR_0 \rfloor + \lfloor wR_1 \rfloor, \tag{9}$$

$$\text{IndexOffset}(R_{1D}) = R_{1D} + \text{argsort}(R_{1D}), \tag{10}$$

where $R_0, R_1 \in [0, 1]$ are the normalized x and y coordinates of the reference location on the image. The argsort function returns the index of each element in the sorted order of $R_{1D}$ in ascending order and is added back to $R_{1D}$ for index reordering. This formulation takes $O(n \log n)$ time, though in practice this operation has negligible cost. The result is a list of one dimensional indices which correspond to the location one column to the left and one row above the hit image location supposing the image features have been flattened using a row major traversal. In cases where other traversals are used, the indices are updated with $\mathcal{T}$. Lastly, from $R_{1D}$ we generate a mask $s_{\text{mask}} \in \mathbb{R}^{(HW+N_{hit})\times 1}$ which marks the insertion points for each query $q_i$ for a hit camera view. To complete the merge operation we copy the queries $Q_{\text{hit}}$ to their corresponding reference points and the values $F_i$ to the masked output tensor. This operation is repeated for every camera and every traversal method. An example of an input sequence can be seen in Figure 4. Ablation studies are also performed examining the traversal order effects in Table 9.

### 3.1.3 CROSS QUASI-SEPARABLE STATE SPACE MODEL

Mamba-2 offered dual methods of computation for selective state space models: as a matrix mixer, SSD and an associative scan, SSM. The reframing of SSMs as structured matrix mixers, $\mathcal{M}$, allows for fast parallel computation by way of batch matrix multiplication and a shortened associative scan. In practice, we utilize SSD during both training and inference as it is better optimized for the hardware used, however we did not adapt the kernel to reflect the true computational complexity of the Cross Quasi-separable State Space Model (XQSSM) module.

The simple sequential implementation in Algorithm 2 (Appendix) shows the unique approach allowed by our module formulation where the discretization factor of the query input, $Q_{dt}$, is set to 0. No hidden state update occurs when a token with $dt = 0$ is processed as shown below

$$h_t = \exp(dt_t A)h_{t-1} + dt_t B_t x_t, \tag{11}$$

$$h_t = \exp(0)h_{t-1} + 0, \tag{12}$$

$$h_t = h_{t-1}. \tag{13}$$

This changes the number of operations per query from $2H(N + 1) + \alpha D(3N + H + 1)$ to $\alpha D(N + H + 1)$. Additionally, outputs from the XQSSM are only needed for query token inputs which reduces the per image feature complexity to $2H(N + 1) + 2\alpha DN$. In total the computational complexity of

the XQSSM is $2V(H(N+1) + \alpha DN) + M\alpha D(N + H + 1)$, where $M$ is the number of queries which are added to the sequence. Additionally, the memory complexity in the sequential form is constant, though when parallelized it becomes linear with respect to the sequence length as shown in Dao & Gu (2024). The resulting matrix mixer, $\mathcal{M}$ goes from a $(M + V \times M + V \times 2H)$ to $(M \times V \times 2H)$ similar to the matrix mixer for dot product cross attention of shape $(Q \times K \times H)$.

### 3.1.4 QUERY AND FEATURE POST-PROCESSING

After the merged input sequence is passed through the inner SSM block, queries must be extracted from the output $Y \in \mathbb{R}^{(HW+M) \times \alpha D}$ to obtain the updated BEV queries $Q'_{BEV}$. The mask generated during the merge operation $s_{\text{mask}}$ is then applied to the output to obtain $Y_Q \in \mathbb{R}^{M \times \alpha D}$. Then using $R_{1D}$ each query in $Y_Q$ is accumulated in $Q_y = 0 \in \mathbb{R}^{N \times \alpha D}$ at its original position in $Q_{BEV}$. Each element $q_y \in Q_y$ has a magnitude which varies depending on the number of queries accumulated in that BEV location. The result is projected to $Q_{out} \in \mathbb{R}^{N \times D}$, then added to the residual BEV query grid $Q$. To account for instability resulting from disparity in magnitude, two methods of normalization are considered: averaging before the out projection and RMS normalization after the out projection. Each method and their joint usage is ablated in Table 7.

**BEV Self-Attention.** The normalized BEV grid Q is traversed with a single hydra layer (bi-directional mamba) in a row-major order before being fed to a deformable layer to incorporate multiscale features.

**Deformable Attention.** Deformable attention is a sparse attention method which excels in detection tasks while training in less time, with fewer flops, and better scaling than vanilla softmax attention Zhu et al. (2020). The foundation of the method is to take an input set of image features and a set of coordinates which act as a reference locations then grid sample the area around the reference location based on using offsets predicted queries.

The computational cost of deformable attention is divided into 3 main components: the cost of calculating the offsets for each query $O(3NMPRD)$, the cost of bilinear interpolation and the weighted sum of samples $O(ND^2 + NPRD^2 + 5NPRD)$, and lastly the cost of computing the linear projection of the values $H_iW_iD^2$ for each image feature level $F_i$ with shape $(H_i \times W_i \times D)$. Here $N$ refers to the number of queries, $D$ is the latent dimension, $P$ is the number of reference points per query, and $R$ is the number of offsets per reference point. The total computational complexity of the operation is $O(ND^2 + \min(HWD^2, NPRD^2) + 5NPRD + 3NMPRD)$ and under the assumption that $5PR + 3MPR < D$, the overall complexity is simplified to $O(2ND^2 + \min(HWD^2, NPRD^2))$. Notably, the computational complexity here is low relative to the size of the image as the heaviest component of the bound derives from the number of queries, heads, and reference points used. In cases where $HW$ is small relative to the number of queries, deformable attention is similar to or worse than linear attention alternatives like Mamba in computational complexity while giving a weaker form of cross attention.

## 4 EXPERIMENTAL SETUP

We follow the methodologies of the previous work of Wang et al. (2022); Li et al. (2022a) and Yang et al. (2023). We evaluate on two backbones: ResNet101 and ResNet50 which are trained in a depth prediction task and COCO, respectively. During training, the first stage of the backbone is frozen, and all other stages are trained at a 10% learning rate to fine-tune their latent representations to the multiview autonomous driving setting.

### 4.1 DATASET AND METRICS

We conduct our experiments using the nuScenes dataset Caesar et al. (2020). The nuScenes dataset is a large-sale autonomous driving dataset containing 1000 driving scenes from Boston and Singapore. Each scene is approximately 20 seconds in duration. 23 object classes with 3D bounding boxes are annotated at 2Hz for the entire dataset, of these 10 are used for the 3D detection task. Each scene is captured using 6 cameras with a FOV of 360 degrees, LiDAR, radar, GPS, sensor calibration, and IMU data. The evaluation metrics and framework for computing them are provided as a part of the nuScenes devkit. The metrics used for evaluation are 1) the mean average precision (mAP), which evaluates both localization and classification performance of the predicted results over four different

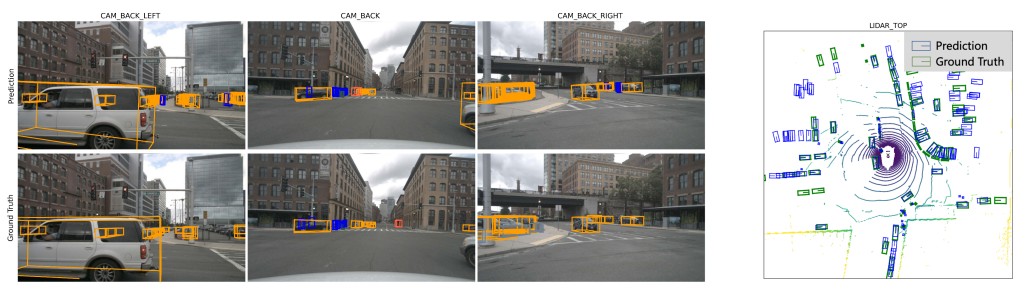

Figure 5: Visualization results of MamBEV-Small on nuScenes val set. We show the 3D bboxes predictions in multi-camera images and the bird's-eye-view.

thresholds using center distance on the ground plane, and 5 types of true positive metrics: 2) average translation error (ATE), 3) average scale error (ASE), 4) average orientation error (AOE), 5) average velocity error (AVE), and 6) average attribute error (AAE). The nuScenes also defines a nuScenes detection score (NDS) by combining mAP with five true positive metrics for a comprehensive assessment. In our method and experiments, only the camera frames, sensor calibration data, and GPS data are used in making predictions.

## 4.2 IMPLEMENTATION DETAILS

We used a learning rate of $8 \times 10^{-4}$, with a linear warmup for 10% of the scheduled steps starting from $\frac{8}{3} \times 10^{-4}$ Following the warmup, the learning rate follows an epoch based cosine annealing schedule with a minimum learning rate of $8 \times 10^{-7}$. We trained with an effective batch size of 32 with no gradient accumulation on 8 A100s for 30 epochs, truncated at 24 epochs. Starting from step 100 an exponential moving average according to the function $w'_t = (1 - 0.0002)w_t + 0.0002w_t$ is applied to all weights. An AdamW optimizer with a 0.01 weight decay is used, and training employs an automatic mixed precision optimizer wrapper with an initial gradient scaling of 512. A 0.1 multiplier is applied to the learning rate of the backbone weights and the deformable attention sampling offsets Zhu et al. (2020). We train the models from scratch using a randomly initialized network for the encoder layers. The source code will be made available upon publication.

## 5 RESULTS

| Method | Backbone | # Frames | NDS↑ | mAP↑ | mATE↓ | mASE↓ | mAOE↓ | mAVE↓ | mAAE↓ |
|---|---|---|---|---|---|---|---|---|---|
| BEVFormerV1-Tiny | ResNet50 | 3 | 0.354 | 0.252 | 0.900 | **0.294** | 0.655 | 0.657 | 0.216 |
| BEVFormerV2-Tiny* | ResNet50 | 3 | 0.397 | **0.270** | 0.820 | 0.301 | 0.594 | 0.469 | **0.195** |
| MamBEV-Tiny | ResNet50 | 3 | **0.399** | 0.266 | **0.794** | 0.298 | **0.575** | 0.469 | 0.199 |
| FCOS3D | ResNet101 | 1 | 0.415 | 0.343 | 0.725 | **0.263** | 0.422 | 1.292 | **0.153** |
| Focal-PETR | ResNet101 | 1 | **0.461** | 0.390 | **0.678** | **0.263** | 0.395 | 0.804 | 0.202 |
| DETR3D | ResNet101 | 1 | 0.434 | 0.349 | 0.716 | 0.268 | **0.379** | 0.842 | 0.200 |
| BEVFormerV1-Small | ResNet101 | 1 | 0.448 | 0.375 | 0.725 | 0.272 | 0.391 | **0.802** | 0.200 |
| BEVFormerV2 | ResNet101 | 1 | 0.426 | 0.355 | 0.751 | 0.275 | 0.429 | 0.847 | 0.215 |
| MamBEV-Small | ResNet101 | 1 | 0.444 | **0.392** | 0.696 | 0.283 | 0.411 | 0.897 | 0.230 |
| PolarDETR-T | ResNet101 | 2 | 0.488 | 0.383 | 0.707 | **0.269** | **0.344** | 0.518 | 0.196 |
| BEVFormerV1-Small | ResNet101 | 3 | 0.479 | 0.370 | 0.721 | 0.279 | 0.407 | 0.436 | 0.220 |
| BEVFormerV1-Base | ResNet101 | 4 | 0.517 | 0.416 | 0.673 | 0.274 | 0.372 | 0.394 | 0.198 |
| MamBEV-Small-Pure† | ResNet101 | 4 | 0.506 | 0.412 | 0.676 | 0.281 | 0.400 | 0.470 | 0.187 |
| MamBEV-Small | ResNet101 | 3 | 0.523 | 0.415 | 0.656 | 0.281 | 0.379 | **0.340** | 0.192 |
| MamBEV-Small | ResNet101 | 4 | **0.525** | **0.423** | 0.662 | 0.280 | 0.386 | 0.354 | **0.183** |

Table 1: Main Results. Our method is the best when accounting for temporal properties. Our model outperforms existing techniques while requiring fewer computational resources. The best results for each experimental setup is highlighted in bold. * indicates models trained by us. † indicates without deformable attention.

We first report our main results followed by ablation studies to understand the efficacy of various model choices. Unless otherwise specified, we used a single temporal frame in all ablation studies. Ablation models were trained for 12 epochs and used a ResNet50 backbone pre-trained on the COCO

object detection dataset. While performance improved with more training, it did not differ relatively to different model parametrizations.

**Main Results.** We present a comparison of our results in Table 1 against state-of-the-art methods at compatible parameter and image input scales. We only use camera features; additionally, we do not make use of any auxiliary loss as in the works of Yang et al. (2023). We aligned the definitions of tiny and small models with BEVFormerV1. We trained only the **tiny** and **small** versions of MamBEV. The CNN based temporal component of our model consists of 57M parameters for the small configuration and 32M parameters for the tiny configuration. Excluding the temporal portion, our MamBEV-Small model has 65M parameters, while MamBEV-Tiny has 39M parameters. For comparison, BEVFormer's tiny, small, and base models have 34, 60, and 69 million parameters respectively.

We report comparable results (+.002 NDS), MamBEV-Tiny, over previous state-of-the-art methods evaluated on similar conditions for the ResNet50 backbone. We report an increase of .046 (+9.6%), MamBEV-Small with 4 frames, in NDS over the previous BEVFormerV1-Small model and 0.008 (1.5%) increase over the previous BEVFormerV1-Base model. This shows that our model can improve performance while reducing memory complexity.

**Effectiveness of Spatial Cross Mamba.** To verify the effectiveness of the Spatial Cross Mamba layer, we train a small model names MamBEV-Small-Pure without deformable layers in the encoder. This configuration displayed good performance in NDS and mAAE metrics that outperforms PolarDETR-T and BEVFormerV1-Small. However, it was outperformed by the small model which made use of deformable layers.

**Efficiency.** In table 2, we test the memory and compute estimated FLOPs for model configurations which use our XQSSM, standard dot product attention, or deformable attention. XQSSM and deformable attention scale linearly in memory and computational complexity with respect to the size of the inputs $V$ and $Q$, though the coefficient factor of deformable attention is smaller.

| Cross Attention Module | BEV Scale (Q) | Image Size (V) | Params (K) | GFLOPs | Memory (GB) |
|---|---|---|---|---|---|
| XQSSM | 50x50 | 800x450 | 239 | 3.7 | 1.7 |
| | 100x100 | 1280x720 | 239 | 14 | 3.5 |
| | 200x200 | 1600x900 | 239 | 51 | 6.5 |
| Deformable | 50x50 | 800x450 | 156 | 3.3 | 1.7 |
| | 100x100 | 1280x720 | 156 | 12.8 | 3.2 |
| | 200x200 | 1600x900 | 156 | 49.5 | 4.9 |
| Dot Product | 50x50 | 800x450 | 263 | 23.9 | 2.2 |
| | 100x100 | 1280x720 | 263 | 228.8 | 9.7 |
| | 200x200 | 1600x900 | 263 | 1,432.5 | >24 |

Table 2: Scaling of cross attention modules with respect to BEV grid and image sizes. All models were tested with a simple R50 backbone and a single encoder and decoder layer. Memory measurements were taken at train time.

**Increasing Temporal Information.** We conduct experiments on the effect of increasing temporal information by providing the model with additional previous frames. Utilizing a higher number of frames requires attending to additional information during the encoding step. Since SSM-based attention can reduce this complexity, we can better utilize this information. We report our results in Table 3.

| # Frames | NDS↑ | mAP↑ | mATE↓ | mASE↓ | mAOE↓ | mAVE↓ | mAAE↓ | Params↓ |
|---|---|---|---|---|---|---|---|---|
| 1 | 0.3128 | 0.2362 | 0.8882 | **0.3098** | 0.7158 | 0.9114 | 0.2269 | 39M |
| 2 | 0.3512 | 0.2415 | 0.8657 | 0.3126 | 0.6743 | 0.6134 | 0.2292 | 53M |
| 3 | 0.3730 | 0.2546 | 0.8480 | 0.3156 | 0.6576 | 0.5073 | **0.2143** | 71M |
| 4 | 0.3747 | 0.2545 | 0.8614 | 0.3132 | 0.6464 | 0.4820 | 0.2227 | 96M |
| 5 | **0.3874** | **0.2705** | **0.8400** | 0.3111 | 0.6323 | **0.4740** | 0.2209 | 128M |
| 8 | 0.3825 | 0.2643 | 0.8372 | 0.3143 | **0.6298** | 0.4971 | 0.2184 | 266M |

Table 3: Performance of our model across all metrics NDS of models on nuScenes validation set using different numbers of temporal frames.

As can be observed, increasing frames helps overall performance until approximately 5 frames. The worst performance occurs when a single frame is used to make predictions. Unsurprisingly, the mAVE, a measurement of velocity prediction error, is also significantly higher when only a single time step of multi-view video is available to the model.

**SSM-based Attention versus Deformable Attention**. We compare our formulation of spatial cross mamba against deformable attention. We present our results in Table 4. Spatial cross mamba can serve as a replacement for deformable attention as there is minimal difference in performance. In our experiments, we found that using mixed spatial and deformable attention was helpful when training a larger network that learns representations over a larger number of frames.

| Method | NDS↑ | mAP↑ | mATE↓ | mASE↓ | mAOE↓ | mAVE↓ | mAAE↓ |
|---|---|---|---|---|---|---|---|
| Deformable Attention | 0.3140 | 0.2384 | 0.8891 | 0.3097 | 0.7166 | **0.9107** | 0.2255 |
| Spatial Cross Mamba | **0.3141** | **0.2386** | 0.8882 | 0.3098 | 0.7158 | 0.9114 | 0.2269 |
| Mixed | 0.3128 | 0.2362 | **0.8799** | **0.3086** | **0.7044** | 0.9349 | **0.2246** |

Table 4: Comparison of encoder layer submodule formulations. Minimal difference is observed between the methods. Mixed attention involves a single layer of spatial cross mamba followed by a single layer of deformable cross attention.

**Scaling Experiments.** We conduct experiments to understand the effect of scaling the number of channels inside the Mamba module as well as the hidden state inside the state space model. We report the hidden state scaling experiments in Table 5 and the channel scaling results in Table 8. Minimal effect is observed in our experiments when adjusting these variables. The findings suggest that increasing the scales of channel features and hidden state may not necessarily lead to improved performance.

| Hidden State | NDS↑ | mAP↑ | mATE↓ | mASE↓ | mAOE↓ | mAVE↓ | mAAE↓ |
|---|---|---|---|---|---|---|---|
| 16 | **0.3167** | 0.2368 | 0.8812 | **0.3073** | **0.7012** | **0.8932** | 0.2339 |
| 32 | 0.3128 | 0.2362 | 0.8882 | 0.3098 | 0.7158 | 0.9114 | **0.2269** |
| 64 | 0.3111 | 0.2356 | 0.8861 | 0.3117 | 0.7015 | 0.9392 | 0.2288 |
| 128 | 0.3125 | 0.2371 | **0.8730** | 0.3106 | 0.7077 | 0.9410 | 0.2287 |
| 256 | 0.3164 | **0.2397** | 0.8790 | 0.3110 | 0.7151 | 0.8979 | 0.2319 |

Table 5: Effect of adjusting SSM hidden state size. Minimal difference in performance is observed.

**Queries Insertion.** To validate our BEV Position Aware Merge (Project), we conduct experiments to show the effectiveness in Table 6. The results indicate that Project is the most effective query insertion method for our model, particularly in terms of detection performance and minimizing key errors. It is reasonable that the performance of append and prepend is worse as Mamba learns spatial relations mainly through the position of elements in a sequence.

| Method | NDS↑ | mAP↑ | mATE↓ | mASE↓ | mAOE↓ | mAVE↓ | mAAE↓ |
|---|---|---|---|---|---|---|---|
| Append | 0.3051 | 0.2314 | 0.8897 | 0.3103 | 0.7560 | 0.9229 | **0.2221** |
| Prepend | 0.3073 | 0.2300 | 0.8917 | 0.3106 | 0.7695 | **0.8887** | 0.2265 |
| Project | **0.3128** | **0.2362** | **0.8882** | **0.3098** | **0.7158** | 0.9114 | 0.2269 |

Table 6: Performance comparison of our model using different query insertion methods. Append/prepend represents naively append/prepend the queries after the corresponding image feature maps. Project is our proposed BEV Position Aware Merge method.

## 6 CONCLUSION

Our work presents a BEV construction model. We show that it improves performance over prior art when evaluated on the similar experimental conditions. Extensive ablation studies demonstrate that the model is robust to various changes. Our source code will be open-source for future research purposes.

## ACKNOWLEDGMENTS

Research was sponsored by the Army Research Laboratory and was accomplished under Cooperative Agreement Number W911NF-23-2-0224 and Toyota Motor North America. The views and conclusions contained in this document are those of the authors and should not be interpreted as representing the official policies, either expressed or implied, of the Army Research Laboratory or the U.S. Government. The U.S. Government is authorized to reproduce and distribute reprints for Government purposes notwithstanding any copyright notation herein. The contents do not necessarily reflect the official views of Toyota Motor North America. This work is partially supported by the National Science Foundation (NSF) under Grant OAC-2017564, which has been instrumental in providing valuable training opportunities for the author(s).

## ETHICS STATEMENT

Our motivation for studying this problem was primarily driven by a desire to minimize the computational costs associated with training and deployment of large deep learning systems. The potential impacts of this are two fold, first it may lead to improved real-time performance properties, second, it would reduce the carbon footprint of training large models.

## REPRODUCIBILITY STATEMENT

Code will be publicly available through github. We will also include installation scripts as the libraries have complex dependencies and compilation needs. We hope this will facilitate faster future development.

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

# A APPENDIX (SUPPLEMENTARY)

## A.1 ADDITIONAL ABLATION

**Feature Normalization.** We show the effect of various norms on the latent BEV queries. We demonstrate this in Table 7. Normalization and averaging features slightly improve the overall performance of the model, as evidenced by the highest NDS and mAP scores. Even without normalization or averaging, the model still performs relatively well, but the combination of these techniques is critical to have the best detection precision.

| Method | NDS↑ | mAP↑ | mATE↓ | mASE↓ | mAOE↓ | mAVE↓ | mAAE↓ |
|---|---|---|---|---|---|---|---|
| Average | 0.3125 | 0.2355 | 0.8886 | **0.3084** | 0.7174 | **0.9056** | 0.2327 |
| RMSNorm | 0.3086 | 0.2358 | 0.8931 | 0.3143 | 0.7337 | 0.9197 | 0.2322 |
| Neither | 0.3110 | 0.2336 | 0.8905 | 0.3092 | **0.6859** | 0.9361 | 0.2367 |
| Both | **0.3128** | **0.2362** | **0.8882** | 0.3098 | 0.7158 | 0.9114 | **0.2269** |

Table 7: Performance comparison of our model using combinations of feature normalization techniques. Using both normalization helped very slightly in performance.

| Expand | NDS↑ | mAP↑ | mATE↓ | mASE↓ | mAOE↓ | mAVE↓ | mAAE↓ |
|---|---|---|---|---|---|---|---|
| 0.5 | 0.3003 | 0.2129 | 0.9041 | **0.2999** | 0.7046 | 0.9255 | **0.2270** |
| 1 | 0.3040 | 0.2177 | 0.9160 | 0.3064 | **0.6842** | 0.9138 | 0.2283 |
| 2 | 0.3036 | 0.2177 | 0.9025 | 0.3044 | 0.6881 | 0.9225 | 0.2350 |
| 4 | **0.3065** | 0.2182 | 0.9064 | 0.3066 | 0.6906 | **0.8771** | 0.2451 |
| 8 | 0.3054 | **0.2216** | **0.8863** | 0.3044 | 0.6862 | 0.9309 | 0.2458 |

Table 8: Comparison of various expansion scales for the linear layer in our Mamba spatial cross mamba. While the performance differences between scales are minimal, the computational cost increases proportionally with the expansion scale.

**Feature Traversal Order.** We examine whether the traversal order on the input features of the spatial cross mamba affects the performance. This is because SSM attention operates on a 1D input sequence but our input is a 2D image feature map. A 2D feature representation is given by the matrix $[[v_{0,0}, ..., v_{0,W}], ..., [v_{H,0}, ..., v_{H,W}]]$. Flattening according to a column-major order changes this to the vector $[v_{0,0}, ..., v_{H,0}, v_{0,1}, ...v_{H,1}, ...]$. Row-major order flattens it to the form $[v_{0,0}, ..., v_{0,W}, v_{1,0}, ...]$. The snake scan changes goes in reverse once it reaches the edge of the matrix. For example, for a horizontal snake scan, the vector would be $[v_{0,0}, ..., v_{0,W}, v_{1,W}, ...v_{1,0}, v_{2,0}, ...]$. We also consider patch based local scans which work by first dividing the image feature map $(H \times W \times D)$ into patches of shape $(H_p \times W_p \times D)$ where $H$ and $W$ are divisible by $H_p$ and $W_p$ respectively. The inner traversal order flattens the resulting $P$ patches into 1D sequences of shape $(H_p W_p \times D)$. The outer traversal determines the order of the flattened patches in the final sequence by mapping the $P$ sequences of length $H_p W_p$ to one sequence with shape $(P H_p W_p \times D)$.

| Traversal Method | NDS↑ |
|---|---|
| Column-major | 0.3176 |
| Row-major | 0.3128 |
| Row-major + column-major | 0.3124 |
| Column snake | 0.3126 |
| Row snake | 0.3201 |
| Row snake + column snake | 0.3188 |
| Row snake inner, column snake outer | 0.3173 |
| Row snake inner, column-major outer | 0.3201 |

Table 9: Effect of Traversal Order.

These patches are internally traversed and the patches are then traversed globally. We also evaluate combinations of scanning order which would lengthen the number of outputs during spatial cross

mamba. Results visualized in Table 9. We found that horizontal snake order is a simple and effective method for scanning and additional traversal orders in a single layer did not improve performance significantly.

| Method | NDS↑ | mAP↑ | mATE↓ | mASE↓ | mAOE↓ | mAVE↓ | mAAE↓ | Params↓ |
|---|---|---|---|---|---|---|---|---|
| Self-Attention | 0.2991 | 0.2196 | 0.9054 | **0.3091** | 0.7498 | 0.9148 | 0.2274 | **37.5M** |
| Deformable | 0.2992 | 0.2246 | 0.9216 | 0.3114 | 0.7437 | **0.9092** | 0.2447 | 37.5M |
| Hydra $\alpha = 2$ | 0.3061 | 0.2282 | 0.8962 | 0.3131 | 0.7167 | 0.9184 | 0.2358 | 38.1M |
| Hydra $\alpha = 4$ | **0.3128** | **0.2362** | **0.8882** | 0.3098 | **0.7158** | 0.9114 | **0.2269** | 38.9M |

Table 10: Performance comparison of different Self-Attention styles.

**BEV Self-Attention Styles.** We conduct experiments to show the effect of various self-attention styles for the BEV query grid. We demonstrate this in Table 10. Along with a slight increase in parameters, the performance of hydra (bi-directional mamba-2) outperforms normal self attention and deformable self attention, as evidenced by the NDS and mAP scores. This validates the effectiveness of our BEV Self-Attention design.

**Inner Cross Mamba Block Order.** We fully explore the merge order and extract order inside the Cross Mamba block. The results are reported in Table 11. Merging after conv1d and extracting queries before gate with $B_Q = 0$, $C_V = 0$, and $dt_Q = 0$ have outstanding performance. While merging before conv1d and extracting queries after gate with $B_Q = 0$ and $dt_Q = 0$ also have not bad performance, this combination suffer from more computation as there are additional conv1d operation for queries.

| Merge Order | Extract Order | Zero Parameters(s) | NDS | mAP |
|---|---|---|---|---|
| Before Conv1D | After Gate | - | 0.2937 | 0.1835 |
| | | $dt_Q$ | 0.2921 | 0.1840 |
| | | $B_Q$ | 0.2991 | 0.1870 |
| | | $B_Q, dt_Q$ | 0.3010 | 0.1870 |
| After Conv1D | Before Gate | - | 0.3035 | 0.1864 |
| | | $dt_Q$ | 0.2978 | 0.1842 |
| | | $B_Q, dt_Q$ | 0.2985 | 0.1892 |
| | | $B_Q, C_V, dt_Q$ | 0.3071 | 0.1946 |
| After Conv1D | After Gate | - | 0.1849 | 0.0614 |
| | | $dt_Q$ | 0.2158 | 0.0895 |

Table 11: Ablation of different merge and extract operation orders tested on a model with 3 layers with alternating layers of SSM Spatial Cross Mamba and SSM Self attention with 2 temporal frames. Other settings of note: self attention uses a hidden state expansion size of 2 instead of 4, with a state dimension of 64 instead of 128 with the same number of attention heads. Additionally, a grouped decoder is used with 6 groups. Trained for 12 epochs on the same schedule as other ablations.

## A.2 Additional Efficiency Analysis

In our assessment, the reported runtime numbers may not accurately reflect the potential of our method, as the implementation we used has not been optimized to the same extent as existing methods. Notably, the authors of Mamba and Mamba-2 have demonstrated that the inner SSM can be computed efficiently and have provided optimized implementations of this block in their works. However, we did not modify this block and instead repurposed the existing Mamba implementation as explained in Section 3.1.3. This highlights the possibility of further refinement and performance improvements in our approach. Compared to attention-based methods, Mamba exhibits significantly better scaling, achieving linear complexity. While the full theoretical efficiency outlined in our work has not yet been fully realized, the estimated FLOPs and inference memory metrics serve as the most accurate representations of our method's performance to date. To underscore our runtime performance gains, we provide a comparative analysis with full attention-based methods in the table 12.

Our primary contribution and goal are not a minor improvement in performance in BEV detection, but rather a method to show how Mamba can be used to achieve that performance while maintaining

| Method | FPS↑ | Memory(GB) ↓ |
|---|---|---|
| Transformer | 3.7 | 9.8 |
| Spatial Cross Mamba | 4.7 | 2.2 |

Table 12: Both models above use 3 layers (cross attention, self attention, ffn), act only on the smallest feature map (23x40), use a 100x100 BEV grid, with a R101 backbone. The FPS is the average number of samples per second processed by the model in evaluation mode on an RTX 4090 GPU. All cameras views are collapsed into a single sequence.

its advantages (linear-cost attention with parallel training). In the table 13, we compare a naïve implementation and our specialized implementation for BEV perception. In this case, the input sequence is relatively short (375 image feature vectors) so a hidden state of size $256 < 375$ is likely to capture all relevant information in the sequence. Compared to our result using projection, method we see a small 0.3% increase in NDS performance while having a per block increase in GFLOPs of 461%, a per block increase in parameters of 98%, and a total memory increase of 10%. The stark difference in efficiency shows the benefit of our work to efficient applications of Mamba for cross attention in BEV and offers direction for future applications to other areas.

| Method | NDS↑ | mAP↑ | Params(K) ↓ (Spatial Cross Mamba) | GFLOPs (K) ↓ (Spatial Cross Mamba) | Memory(MB) ↓ (Total) |
|---|---|---|---|---|---|
| Append h=32 | 0.3051 | 0.2314 | 240 | 2.6 | 438 |
| Append h=256 | 0.3138 | 0.2371 | 476 | 14.6 | 480 |
| Project h=32 | 0.3128 | 0.2362 | 240 | 2.6 | 438 |

Table 13: Experiments used a single temporal frame and were trained for 12 epochs and used a ResNet50 backbone pre-trained on the COCO object detection dataset.

### A.3   ALGORITHMS

The Pseudocode of our proposed Spatial Cross Mamba shows in Algorithm 1. The details of the Cross Quasi-Separable State Space Model (XQSSM) show in Algorithm 2.

---

**Algorithm 1** Spatial Cross Mamba

---

**Input:**   $q : (B, Q, D)$, $v : (B, C, V, D)$, $r : (B, C, Q, Z, 2)$
**Output:**   $y : (B, Q, D)$
 1: $A : (H,) \leftarrow$ Parameter
 2: $b : (B, C, Q, Z) \leftarrow 0 \leq r \leq 1$                     ▷ Mask invalid reference points
 3: $M : (1, ) \leftarrow \text{sum}(b)$                                    ▷ ≈ZQ
 4: $Q_{zxBC} : (B, Q, 2\alpha D+4NG) \leftarrow \text{MaskedLinear}_{in}(q, dt)$
 5: $V_{xBCdt} : (B, C, V, \alpha D+4NG+2H) \leftarrow \text{MaskedLinear}_{in}(v, z)$
 6: $V_{xBCdt} : (BCV, \alpha D) \leftarrow \text{flatten}(V_{xBCdt}, [0,1,2])$
 7: $Q_z : (B, Q, \alpha D)$, $Q_{xBC} : (B, Q, \alpha D+4NG) \leftarrow \text{split}(Q_{zxBC})$
 8: $Q_{xBCdt} : (B, Q, \alpha D+4NG+2H) \leftarrow \text{concat}([Q_{xBC}, \text{repeat}(0, 2H)])$
 9: $xBCdt : (M+BCV, \alpha D+4NG+2H)$, extract $: (M)$, $s_{\text{mask}} : (M+BCV)$
        $\leftarrow \text{Merge}(Q_{xBCdt}, V_{xBCdt}, b, r)$
10: $y_Q : (M+BCV, \alpha D) \leftarrow \text{XQSSM}(xBCdt, A)$
11: $y_{\text{gated}} : (M, \alpha D) \leftarrow \text{RMSGated}(y_Q[s_{\text{mask}}], Q_z)$
12: $Q_y : (B, Q, \alpha D) \leftarrow \text{indexAdd}(y_Q, \text{extract})$
13: count $: (B, Q) \leftarrow \text{clamp}(\text{sum}(b, [1, 3]), 1.0)$
14: $Q_{\text{avg}} (B, Q, \alpha D) \leftarrow Q_y/\text{count}$
15: $Q_{\text{out}} : (B, Q, D) \leftarrow \text{FusedNormDropResidual}(\text{Linear}_{out}(Q_{\text{avg}}), q)$

---

---

**Algorithm 2** Cross Quasi-Separable State Space Model (XQSSM) – Recurrent Form

---

**Input:** $x$ : (2, M+BCV, $\alpha$D), $B$ : (2, M+BCV, N), $C$ : (2, M+BCV, N), $A$:(H), $dt$:(2, M+BCV, H), $s_{\text{mask}}$ : (2, M+BCV)

**Output:** $y$ : (M, $\alpha$D)

1:   $dt_{\text{bias}}$ : (2H) $\leftarrow$ Parameter
2:   $\Delta$ : (2, BCV, H) $\leftarrow$ SoftPlus($dt[s_{\text{mask}}] + dt_{\text{bias}}$) $\triangleright$ $s_{\text{mask}} = 1$ where $x_{ij}$ is an image feature vector
3:   $dA$ : (2, BCV, H) $\leftarrow$ exp($\Delta A[s_{\text{mask}}]$)
4:   $dBx$ : (2, BCV, $\alpha$ND) $\leftarrow$ $\Delta Bx[s_{\text{mask}}]$
5:   $h$ : (2, H, $\alpha$D//H, N) $\leftarrow$ 0
6:   **for** $i = 0, \dots, 1$ **do**                                                                     $\triangleright$ Forward and backward scans
7:       idx $\leftarrow$ 0
8:       **for** $j = 0, \dots$,M+BCV **do**
9:           **if** $s_{ij}$ **then**
10:               $h_i \leftarrow dA_{ij}h_i + dBx_{ij}$                                         $\triangleright$ Only update state if $x_{ij} \in V$
11:           **else**
12:               $y_{i,\text{idx}} \leftarrow C_{ij}h_i$                                             $\triangleright$ Only compute output if $x_{ij} \in Q$
13:               idx $\leftarrow$ idx + 1
14:           **end if**
15:       **end for**
16: **end for**

---

## A.4   VISUALIZATION

As shown in Figure 6 and 7, we compare MamBEV-Small with BEVFormer-Base. MamBEV-Small successfully detects cars occluded by obstructions with a higher degree of accuracy compared to BEVFormer-Base. The predictions of MamBEV-Small exhibit a closer alignment with the ground truth in terms of both spatial positioning and bounding box dimensions. We also provide more detection results in Figure 8, 9, where our model can successfully detect fully occluded vehicles as well as small and distant objects.

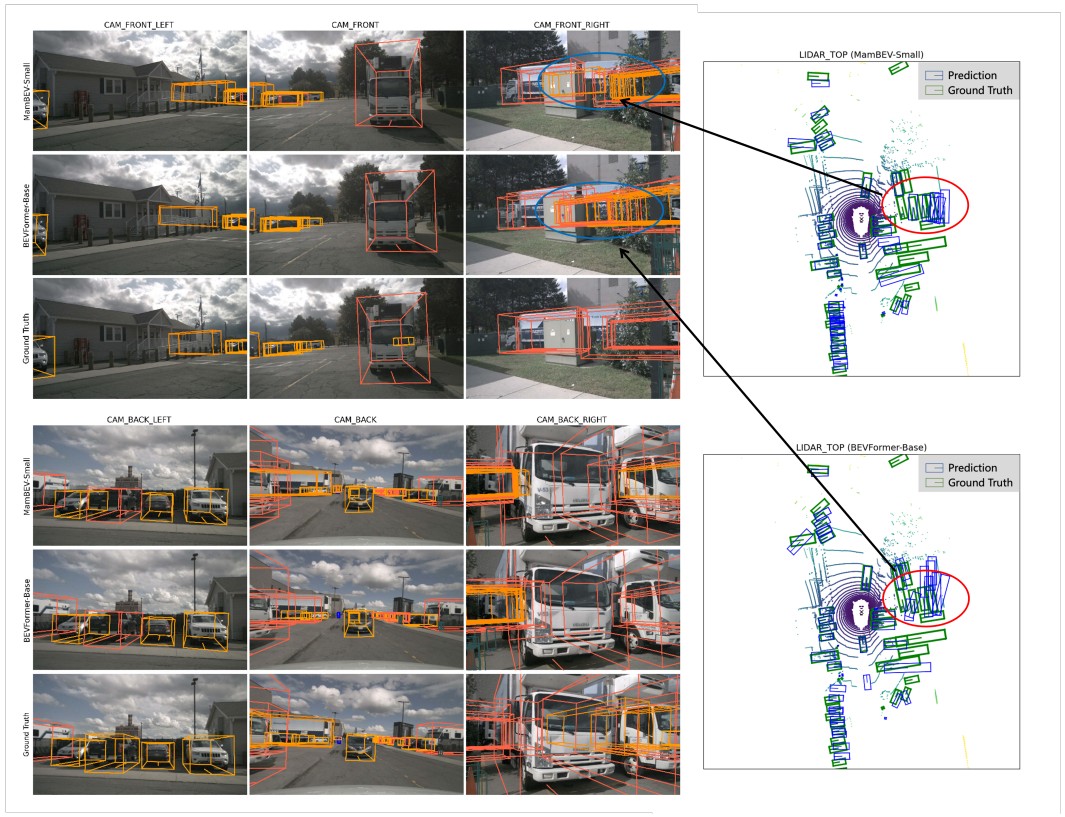

Figure 6: Comparison of MamBEV-Small and BEVFormer-Base on nuScenes val set. We observe that our model can detect highly occluded cars with a higher degree of accuracy compared to BEVFormer-Base.

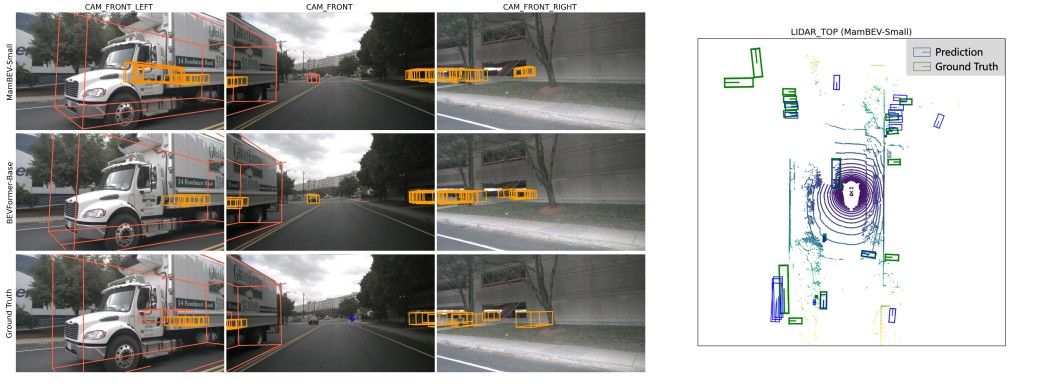

Figure 7: Comparison of MamBEV-Small and BEVFormer-Base on nuScenes val set. We observe that our model can successfully detect objects are missed in the prediction results of BEVFormer-Base.

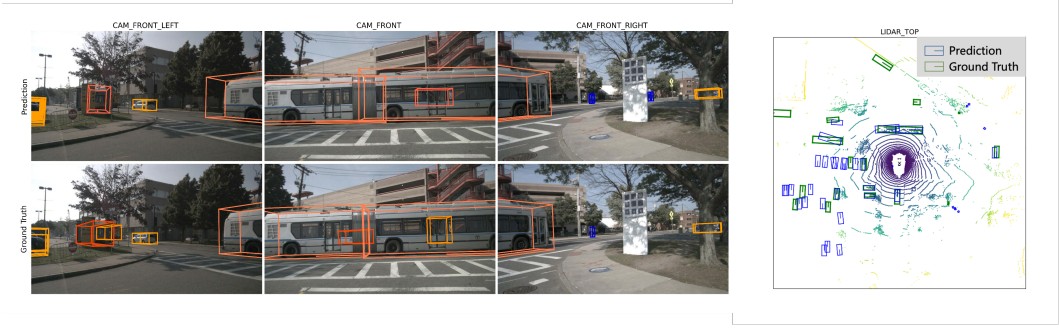

Figure 8: Visualization results of MamBEV-Small on nuScenes val set. We observe that our model can detect highly occluded objects.

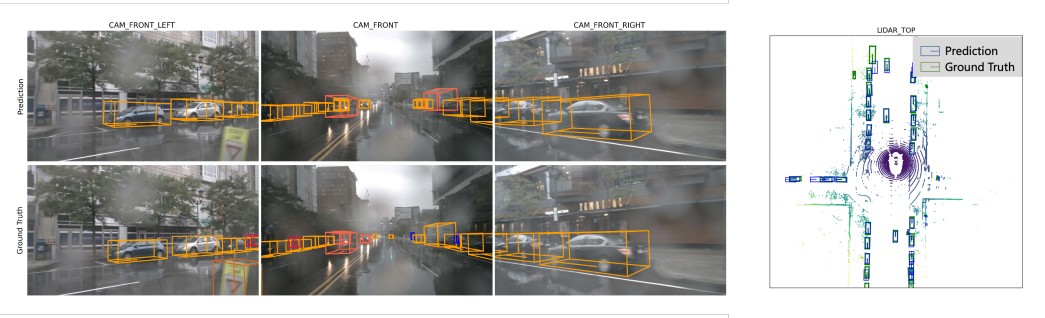

Figure 9: Visualization results of MamBEV-Small on nuScenes val set. We can see that our model can detect objects that are tiny and far away.

