# OpenReview forum: "MamBEV: Enabling State Space Models to Learn Birds-Eye-View Representations"
_ICLR.cc/2025/Conference — ICLR 2025 Poster_

### Official Review · Reviewer_yyMe · 2024-10-19

**Soundness:** 2
**Presentation:** 2
**Contribution:** 2
**Rating:** 6
**Confidence:** 4

**Summary:**

The paper presents MamBEV, a novel framework designed for 3D visual perception tasks like 3D detection from multi-camera images, which are crucial for autonomous driving and assistance systems. The framework leverages a Mamba-based approach to learn unified Bird’s Eye View (BEV) representations using linear spatio-temporal SSM-based attention, which improves computational and memory efficiency across multiple 3D perception tasks. The key is SSM-based cross-attention, which functions similarly to standard cross-attention, enabling BEV query representations to interact effectively with relevant image features. Experiments are conducted to demonstrate the effectiveness of the method.

Pos:
- Enabling Mamba to BEV representation seems to be interesting.

Cons:
- The authors claim that proposed method supports multiple perceptual tasks but has no proof.
- The article keeps emphasizing computational efficiency, but without proofs.
- The core figures are crude and difficult to understand
- Insufficient experimentation and comparison methods

**Strengths:**

- Enabling Mamba to BEV representation seems to be interesting.

**Weaknesses:**

Cons:
- The authors claim that proposed method supports multiple perceptual tasks but has no proof.
- The article keeps emphasizing computational efficiency, but without proofs.
- The core figures are crude and difficult to understand
- Insufficient experimentation and comparison methods

**Questions:**

Please refer to the above.

---

> ### Author Response · Authors · 2024-11-30
> **Response to reviewer yyMe**
>
> Thank you for your thoughtful review, valuable feedback, and the time taken to provide constructive feedback to strengthen our work further! We are pleased that you recognized the efficiency and novel design of our MamBEV. We will address your concerns and questions below:
>
> > **"The authors claim that proposed method supports multiple perceptual tasks but has no proof."**
>
> Thank you for the question. Our primary focus was not on multiple perceptual tasks however we do reference that BEV representations in prior works have been shown to be useful in multiple tasks such as map segmentation and occupancy prediction as they provide a dense representation of a scene. The main goal of our paper is to create a Cross Attention mechanism to SSMs, and show the effectiveness and efficiency of our proposed Spatial Cross Mamba Block.

---

> ### Author Response · Authors · 2024-11-30
> **Respnse to reviewer yyMe**
>
> > **"The article keeps emphasizing computational efficiency, but without proofs."**
>
> Mamba-2 offers **dual methods of computation** for selective state space models:
>
> 1. **Matrix Mixer (SSD)**: Implements structured matrix mixers for fast, parallel computation using batch matrix multiplication.
> 2. **Associative Scan (SSM)**: Allows efficient sequential computation through shortened associative scans.
>
> The reframing of SSMs as structured matrix mixers, denoted as \($\mathcal{M}$), provides enhanced computational efficiency through:
> - **Parallelism**: Enabled by batch matrix multiplication.
> - **Flexibility**: Simplified operations with structured matrix formulations.
>
> In practice, we utilize **SSD** during both training and inference because it is better optimized for the hardware used. However, the **kernel** was not adapted to reflect the true computational complexity of the **Cross Quasi-Separable State Space Model (XQSSM)** module.
>
> ---
>
> ### Sequential Implementation and Query Discretization
>
> Algorithm 2 (refer to Appendix) outlines a **sequential implementation** leveraging the unique formulation of our module. This includes **query discretization**, where the query input \(Q_{dt}\) has a discretization factor of \(dt = 0\). In this case, **no hidden state update occurs**, as demonstrated below:
>
> $\begin{equation}
> h_t = \exp(dt_t A) h_{t-1} + dt_t B_t x_t,
> \end{equation}$
>
> $\begin{equation}
> h_t = \exp(0) h_{t-1} + 0,
> \end{equation}$
>
> $\begin{equation}
> h_t = h_{t-1}.
> \end{equation}$
>
> When \(dt = 0\), this reduces the number of operations per query from:
>
> 2H(N+1) + $\alpha$ D(3N + H + 1)
>
> to:
>
> $\alpha$ D(N + H + 1).
>
> ---
>
> ### Computational Complexity of XQSSM
>
> Outputs from the XQSSM are only required for **query token inputs**, further reducing the **per-image feature complexity** to:
>
> 2H(N+1) + 2$\alpha$ DN.
>
> The **total computational complexity** of the XQSSM is:
>
> 2V (H(N+1) + $\alpha$ DN ) + M $\alpha$ D(N + H + 1),
>
> where:
> - \(M\): Number of queries added to the sequence,
> - \(V\): Number of input features,
> - \(H\): Hidden state size,
> - \(N\): Input sequence length,
> - \($\alpha$ D\): Complexity scaling factor.
>
> The **memory complexity** of the sequential form is **constant**, while in parallelized form, it becomes **linear** with respect to the sequence length, as shown in [1].
>
> ---
>
> ### Matrix Mixer Comparison
>
> The resulting matrix mixer, \($\mathcal{M}\$), reduces in dimensionality from:
>
>
> (M+V) $\times$ (M+V) $\times$ 2H
>
>
> to:
>
>
> M $\times$ V $\times$v2H,
>
>
> which is comparable to the matrix mixer for **dot product cross attention** with a shape of:
>
>
> Q $\times$ K $\times$ H.
>
>
> In the following table, we test the memory and compute estimated FLOPs for model configurations which use our XQSSM, standard dot product attention, or deformable attention. XQSSM and deformable attention scale linearly in memory and computational complexity with respect to the size of the inputs $V$ and $Q$, though the coefficient factor of deformable attention is smaller.
>
> | **Cross Attention Module** | **BEV Scale (Q)** | **Image Size (V)** | **Params (K)** | **GFLOPs** | **Memory (GB)** |
> |-----------------------------|-------------------|--------------------|----------------|-------------|-----------------|
> |                             | 50x50            | 800x450           | 239            | 3.7         | 1.7             |
> | **XQSSM**                  | 100x100          | 1280x720          | 239            | 14.0        | 3.5             |
> |                             | 200x200          | 1600x900          | 239            | 51.0        | 6.5             |
> |-----------------------------|-------------------|--------------------|----------------|-------------|-----------------|
> |                             | 50x50            | 800x450           | 156            | 3.3         | 1.7             |
> | **Deformable**             | 100x100          | 1280x720          | 156            | 12.8        | 3.2             |
> |                             | 200x200          | 1600x900          | 156            | 49.5        | 4.9             |
> |-----------------------------|-------------------|--------------------|----------------|-------------|-----------------|
> |                             | 50x50            | 800x450           | 263            | 23.9        | 2.2             |
> | **Dot Product**            | 100x100          | 1280x720          | 263            | 228.8       | 9.7             |
> |                             | 200x200          | 1600x900          | 263            | 1,432.5     | >24             |
>
> ---
> [1] Tri Dao and Albert Gu. Transformers are ssms: Generalized models and efficient algorithms through structured state space duality. arXiv preprint arXiv:2405.21060, 2024.
>
> We believe this design significantly improves the computational and memory efficiency of the XQSSM module, aligning with the broader goals of efficient 3D spatial awareness in Mamba-2. Thank you again for the opportunity to refine our explanations!

---

> ### Author Response · Authors · 2024-12-01
> **Response to reviewer yyMe**
>
> > **"The core figures are crude and difficult to understand."**
>
>  We have revised our paper, and have a better description of our proposed modules.
>
>
>
>
> > **"Insufficient experimentation and comparison methods"**
>
>
>
> Thank you for your question. The main goal of this paper is to bring the cross attention mechanism to mamba, so that we propose our novel Spatial Cross Mamba. It is important to note that we only trained tiny and small models with scaling consistent with Bevformer's scaling. However, our small version of the model, MamBEV-Small, already outperforms Bevformer-Base. We have had extra ablation experiments to show the effectiveness of our model and dissect the contributions of each component of our model.
>
> ### 1. Performance Comparison of Self-Attention Styles
>
> We conduct experiments to show the effect of various self-attention styles for the BEV query grid. The results are summarized in the table above. Along with a slight increase in parameters, the performance of Hydra (bi-directional Mamba-2) outperforms both standard self-attention and deformable self-attention. This is evidenced by the improved **NDS** and **mAP** scores, which validate the effectiveness of our BEV Self-Attention design.
>
> | **Method**        | **NDS $\uparrow$** | **mAP $\uparrow$** | **mATE $\downarrow$** | **mASE $\downarrow$** | **mAOE $\downarrow$** | **mAVE $\downarrow$** | **mAAE $\downarrow$** | **Params $\downarrow$** |
> |--------------------|--------------------|--------------------|-----------------------|-----------------------|-----------------------|-----------------------|-----------------------|-------------------------|
> | **Self-Attention** | 0.2991            | 0.2196            | 0.9054               | **0.3091**            | 0.7498               | 0.9148               | 0.2274               | **37.5M**              |
> | **Deformable**     | 0.2992            | 0.2246            | 0.9216               | 0.3114               | 0.7437               | **0.9092**           | 0.2447               | 37.5M                  |
> | **Hydra $\alpha=2$** | 0.3061           | 0.2282            | 0.8962               | 0.3131               | 0.7167               | 0.9184               | 0.2358               | 38.1M                  |
> | **Hydra $\alpha=4$** | **0.3128**       | **0.2362**        | **0.8882**           | 0.3098               | **0.7158**           | 0.9114               | **0.2269**           | 38.9M                  |
>
> ---
>
> ### 2. The effectiveness of Spatial Cross Mamba
>
> This table compares encoder layer submodule formulations. Minimal differences are observed between the methods. **Mixed attention** involves a single layer of spatial cross Mamba followed by a single layer of deformable cross attention. While **Spatial Cross Mamba** achieves the highest **NDS** and **mAP**, the **Mixed** method shows improvements in **mATE**, **mASE**, and **mAAE**, highlighting the trade-offs between these approaches. Spatial cross mamba can serve as a replacement for deformable attention in BEV encoding as there is minimal difference in performance. In our experiments, we found that using mixed spatial and deformable attention was helpful when training a larger network that learns representations over a larger number of frames.
>
>
> | **Method**               | **NDS $\uparrow$** | **mAP $\uparrow$** | **mATE $\downarrow$** | **mASE $\downarrow$** | **mAOE $\downarrow$** | **mAVE $\downarrow$** | **mAAE $\downarrow$** |
> |--------------------------|--------------------|--------------------|-----------------------|-----------------------|-----------------------|-----------------------|-----------------------|
> | **Deformable Attention** | 0.3140            | 0.2384            | 0.8891               | 0.3097               | 0.7166               | **0.9107**           | 0.2255               |
> | **Spatial Cross Mamba**  | **0.3141**        | **0.2386**        | 0.8882               | 0.3098               | 0.7158               | 0.9114               | 0.2269               |
> | **Mixed**                | 0.3128            | 0.2362            | **0.8799**           | **0.3086**           | **0.7044**           | 0.9349               | **0.2246**           |
>
> ---
>
> **<CONTINUED>**

---

> > ### Author Response · Authors · 2024-12-01
> > **Response to reviewer yyMe**
> >
> > ### 3. Performance Comparison of Query Insertion Methods
> >
> > This table presents a performance comparison of our model using different query insertion methods:
> > - **Append** and **Prepend** represent naive methods of appending/prepending the queries after the corresponding image feature maps.
> > - **Project** is our proposed **BEV Position Aware Merge** method.
> >
> > The **Project** method consistently achieves the best performance across key metrics such as **NDS**, **mAP**, **mATE**, **mASE**, and **mAOE**, demonstrating the effectiveness of our approach. However, **Append** achieves the lowest **mAAE**, and **Prepend** results in the lowest **mAVE**.
> >
> > | **Method**  | **NDS $\uparrow$** | **mAP $\uparrow$** | **mATE $\downarrow$** | **mASE $\downarrow$** | **mAOE $\downarrow$** | **mAVE $\downarrow$** | **mAAE $\downarrow$** |
> > |-------------|--------------------|--------------------|-----------------------|-----------------------|-----------------------|-----------------------|-----------------------|
> > | **Append**  | 0.3051            | 0.2314            | 0.8897               | 0.3103               | 0.7560               | 0.9229               | **0.2221**           |
> > | **Prepend** | 0.3073            | 0.2300            | 0.8917               | 0.3106               | 0.7695               | **0.8887**           | 0.2265               |
> > | **Project** | **0.3128**        | **0.2362**        | **0.8882**           | **0.3098**           | **0.7158**           | 0.9114               | 0.2269               |
> >
> > ---

---

> ### Comment · Area_Chair_RHLe · 2024-12-01
>
> Dear Reviewer,
>
> Could you look at the authors rebuttal and see if your concerns have been addressed?
>
> Thanks

---

> > ### Comment · Reviewer_yyMe · 2024-12-02
> >
> > Thanks for your detailed reply. I have updated my score.

---

### Official Review · Reviewer_4oR6 · 2024-10-23

**Soundness:** 4
**Presentation:** 3
**Contribution:** 3
**Rating:** 8
**Confidence:** 3

**Summary:**

The paper introduces the Mamba mechanism into 3D visual perception task to learn bird's-eye view (BEV) representations in order to address computational and memory efficiency problems, and designs a Spatial Cross Attention module for the task. The effectiveness of the design is demonstrated through extensive experiments and ablation studies, which would bring benefits to application scenarios such as autonomous driving and driver assistance systems.

**Strengths:**

1. MamBEV proposes a design based on SSM, which exceeds the performance of existing Transformer-based structures in several metrics.
2. MamBEV provides sufficient experimental results to illustrate the effectiveness of the design.

**Weaknesses:**

1. Lack of visualization of results. I am quite sorry to say that I am not familiar with the nuScenes dataset, so could the authors provide some visual examples to show the advantages of MamBEV?
2. Lack of more detailed description of the effectiveness of the Spatial Cross Attention module. Since the authors mention in the Introduction that "we further demonstrate that our method can better capture longer dependencies in multiview video", whether this word "better" means the module proposed in the paper comparing to other SSM Attention designs, or Mamba-2 comparing to Mamba-1, or Mamba comparing to Transformer, could it be supported and presented through other ways in addition to the accuracy results on the specific dataset?

**Questions:**

The main issues have been raised in Weaknesses section.

---

> ### Author Response · Authors · 2024-11-30
> **Response to reviewer 4oR6**
>
> We truly appreciate your thorough review, positive feedback, and time and efforts taken to help us strengthen the paper even more! We are thrilled that you recognized the effectiveness of MamBEV. We are especially pleased that you appreciated our comprehensive evaluation. We will address your concerns and questions below:
>
> > **"Lack of visualization of results."**
>
> Thank you for your question! We have included five detection visualiztion results in our revised paper, particularly in Figure 5, 6, 7, 8, 9.
> In Figure 6 and 7, we compare MamBEV-Small with BEVFormer-Base. MamBEV-Small successfully detects cars occluded by obstructions with a higher degree of accuracy compared to BEVFormer-Base. The predictions of MamBEV-Small exhibit a closer alignment with the ground truth in terms of both spatial positioning and bounding box dimensions. We also provide more detection results in Figure 5, 8, 9, where our model can successfully detect fully occluded vehicles as well as small and distant objects.

---

> ### Author Response · Authors · 2024-12-01
> **Response to reviewer 4oR6**
>
> Thank you for your question! We show the effectiveness of our proposed Spatial Cross Mamba including comparing with deformable attention:
>
> ### The effectiveness of Spatial Cross Mamba
>
> This table compares encoder layer submodule formulations. Minimal differences are observed between the methods. **Mixed attention** involves a single layer of spatial cross Mamba followed by a single layer of deformable cross attention. While **Spatial Cross Mamba** achieves the highest **NDS** and **mAP**, the **Mixed** method shows improvements in **mATE**, **mASE**, and **mAAE**, highlighting the trade-offs between these approaches. Spatial cross mamba can serve as a replacement for deformable attention in BEV encoding as there is minimal difference in performance. In our experiments, we found that using mixed spatial and deformable attention was helpful when training a larger network that learns representations over a larger number of frames.
>
>
> | **Method**               | **NDS $\uparrow$** | **mAP $\uparrow$** | **mATE $\downarrow$** | **mASE $\downarrow$** | **mAOE $\downarrow$** | **mAVE $\downarrow$** | **mAAE $\downarrow$** |
> |--------------------------|--------------------|--------------------|-----------------------|-----------------------|-----------------------|-----------------------|-----------------------|
> | **Deformable Attention** | 0.3140            | 0.2384            | 0.8891               | 0.3097               | 0.7166               | **0.9107**           | 0.2255               |
> | **Spatial Cross Mamba**  | **0.3141**        | **0.2386**        | 0.8882               | 0.3098               | 0.7158               | 0.9114               | 0.2269               |
> | **Mixed**                | 0.3128            | 0.2362            | **0.8799**           | **0.3086**           | **0.7044**           | 0.9349               | **0.2246**           |
>
> ---
>
>
> We also dissect the contributions of each component of our model:
>
>
> ### 1. Performance Comparison of Self-Attention Styles
>
> We conduct experiments to show the effect of various self-attention styles for the BEV query grid. The results are summarized in the table above. Along with a slight increase in parameters, the performance of Hydra (bi-directional Mamba-2) outperforms both standard self-attention and deformable self-attention. This is evidenced by the improved **NDS** and **mAP** scores, which validate the effectiveness of our BEV Self-Attention design.
>
>
> | **Method**        | **NDS $\uparrow$** | **mAP $\uparrow$** | **mATE $\downarrow$** | **mASE $\downarrow$** | **mAOE $\downarrow$** | **mAVE $\downarrow$** | **mAAE $\downarrow$** | **Params $\downarrow$** |
> |--------------------|--------------------|--------------------|-----------------------|-----------------------|-----------------------|-----------------------|-----------------------|-------------------------|
> | **Self-Attention** | 0.2991            | 0.2196            | 0.9054               | **0.3091**            | 0.7498               | 0.9148               | 0.2274               | **37.5M**              |
> | **Deformable**     | 0.2992            | 0.2246            | 0.9216               | 0.3114               | 0.7437               | **0.9092**           | 0.2447               | 37.5M                  |
> | **Hydra $\alpha=2$** | 0.3061           | 0.2282            | 0.8962               | 0.3131               | 0.7167               | 0.9184               | 0.2358               | 38.1M                  |
> | **Hydra $\alpha=4$** | **0.3128**       | **0.2362**        | **0.8882**           | 0.3098               | **0.7158**           | 0.9114               | **0.2269**           | 38.9M                  |
>
> ---
>
> ### 2.  Performance Comparison of Merge and Extract Operation Orders
>
> We fully explore the merge order and extract order inside the Cross Mamba block. The results are reported in Table \ref{tab:my_label}. Merging after conv1d and extracting queries before gate with $B_Q = 0$, $C_V = 0$, and $dt_Q = 0$ have outstanding performance. While merging before conv1d and extracting queries after gate with $B_Q = 0$ and $dt_Q = 0$ also have not bad performance, this combination suffers from more computation as there are additional conv1d operations for queries.
>
> **<CONTINUED>**

---

> ### Author Response · Authors · 2024-12-01
> **Response to reviewer 4oR6**
>
> | **Merge Order**        | **Extract Order**  | **Zero Parameters (s)**       | **NDS**   | **mAP**   |
> |-------------------------|--------------------|--------------------------------|-----------|-----------|
> |                         |                    | \(-\)                          | 0.2937    | 0.1835    |
> |                         |                    | \(dt_Q\)                       | 0.2921    | 0.1840    |
> | **Before Conv1D**       | **After Gate**     | \(B_Q\)                        | 0.2991    | 0.1870    |
> |                         |                    | \(B_Q, dt_Q\)                  | 0.3010    | 0.1870    |
> |-------------------------|--------------------|--------------------------------|-----------|-----------|
> |                         |                    | \(-\)                          | 0.3035    | 0.1864    |
> |                         |                    | \(dt_Q\)                       | 0.2978    | 0.1842    |
> | **After Conv1D**        | **Before Gate**    | \(B_Q, dt_Q\)                  | 0.2985    | 0.1892    |
> |                         |                    | \(B_Q, C_V, dt_Q\)             | 0.3071 | 0.1946 |
> |-------------------------|--------------------|--------------------------------|-----------|-----------|
> |                         |                    | \(-\)                          | 0.1849    | 0.0614    |
> | **After Conv1D**        | **After Gate**     | \(dt_Q\)                       | 0.2158    | 0.0895    |
>
> ---
>
>
>  ### 3. Performance Comparison of Query Insertion Methods
>
> This table presents a performance comparison of our model using different query insertion methods:
> - **Append** and **Prepend** represent naive methods of appending/prepending the queries after the corresponding image feature maps.
> - **Project** is our proposed **BEV Position Aware Merge** method.
>
> The **Project** method consistently achieves the best performance across key metrics such as **NDS**, **mAP**, **mATE**, **mASE**, and **mAOE**, demonstrating the effectiveness of our approach. However, **Append** achieves the lowest **mAAE**, and **Prepend** results in the lowest **mAVE**.
>
> | **Method**  | **NDS $\uparrow$** | **mAP $\uparrow$** | **mATE $\downarrow$** | **mASE $\downarrow$** | **mAOE $\downarrow$** | **mAVE $\downarrow$** | **mAAE $\downarrow$** |
> |-------------|--------------------|--------------------|-----------------------|-----------------------|-----------------------|-----------------------|-----------------------|
> | **Append**  | 0.3051            | 0.2314            | 0.8897               | 0.3103               | 0.7560               | 0.9229               | **0.2221**           |
> | **Prepend** | 0.3073            | 0.2300            | 0.8917               | 0.3106               | 0.7695               | **0.8887**           | 0.2265               |
> | **Project** | **0.3128**        | **0.2362**        | **0.8882**           | **0.3098**           | **0.7158**           | 0.9114               | 0.2269               |
>
> ---

---

> > ### Comment · Reviewer_4oR6 · 2024-12-01
> > **Why can Spatial Cross Attention module better capture longer dependencies**
> >
> > Thanks for your response, especially more detailed performance comparison. However, I still have trouble finding the reason **why** or getting the way **how** Spatial Cross Attention module can better capture longer dependencies in multiview video. Perhaps I need your deeper explanation.

---

> > > ### Author Response · Authors · 2024-12-02
> > > **Response to reviewer 4oR6**
> > >
> > > Thank you for you question!
> > >
> > > In our work, "long dependencies" refer to the extensive sequence length of inputs processed by the encoder. This includes managing large volumes of data from both image feature vectors (14,720 per camera) and BEV queries (10,000 in total for MamBEV-Small). Transformers can handle interactions at a scale of $( (14,720 \times 6) \times 10,000 )$, which poses significant memory challenges, as detailed in following table. Mamba offers a more memory-efficient alternative but exhibits faster hidden state decay, particularly in densely packed sequences. Naive implementations attempt to address these complexities but often trade-off memory and compute efficiency. In the BEV setting, each query corresponds to a specific location on the grid, representing a pillar in real-world space, with relevant information distributed vertically along its 2D projection onto the image. When flattened into a 1D sequence for MamBEV processing, points along the pillar may appear widely separated, lacking a singular location to capture all relevant information effectively. Our proposed solution involves inserting fewer copies (Z << T) of each query along the pillar, complemented by smaller hidden states (N << T), aiming to optimize information capture amid long dependencies between queries and image data.
> > >
> > >
> > > | **Cross Attention Module** | **BEV Scale (Q)** | **Image Size (V)** | **Params (K)** | **GFLOPs** | **Memory (GB)** |
> > > |-----------------------------|-------------------|--------------------|----------------|-------------|-----------------|
> > > |                             | 50x50            | 800x450           | 239            | 3.7         | 1.7             |
> > > | **XQSSM**                  | 100x100          | 1280x720          | 239            | 14.0        | 3.5             |
> > > |                             | 200x200          | 1600x900          | 239            | 51.0        | 6.5             |
> > > |-----------------------------|-------------------|--------------------|----------------|-------------|-----------------|
> > > |                             | 50x50            | 800x450           | 156            | 3.3         | 1.7             |
> > > | **Deformable**             | 100x100          | 1280x720          | 156            | 12.8        | 3.2             |
> > > |                             | 200x200          | 1600x900          | 156            | 49.5        | 4.9             |
> > > |-----------------------------|-------------------|--------------------|----------------|-------------|-----------------|
> > > |                             | 50x50            | 800x450           | 263            | 23.9        | 2.2             |
> > > | **Dot Product**            | 100x100          | 1280x720          | 263            | 228.8       | 9.7             |
> > > |                             | 200x200          | 1600x900          | 263            | 1,432.5     | >24             |
> > > ---

---

> > > > ### Comment · Reviewer_4oR6 · 2024-12-02
> > > > **A clear explanation**
> > > >
> > > > Thank you for your clear explanation! I'm happy to increase my rating, and I hope some details from the rebuttal response can be added to the paper if your work is accepted by the conference.

---

> ### Comment · Reviewer_4oR6 · 2024-12-01
> **Weakness 1 has been solved**
>
> Thanks for your response. My first point in Weakness section has been well explained by your response.

---

### Official Review · Reviewer_yMTq · 2024-11-01

**Soundness:** 4
**Presentation:** 4
**Contribution:** 3
**Rating:** 8
**Confidence:** 3

**Summary:**

The paper proposes a new approach to building bird's-eye-view (BEV) maps from multiple cameras. With an aim of mobile deployment, the paper investigates approaches that depart from the existing transformer-based push (from images to 3D) or pull (from 3d to images) methods. In particular, recent work on state space models (SSM) is an inspiration to replace transformers and their quadratic attention computational complexity. Of particular interest here is to leverage ideas from the Mamba-2 and Hydra SSMs and adapt them to the BEV problem. The new SSM-based BEV model is well described and well tested, including several ablation studies.

Two potential caveats noted below are that only one dataset is used (as seems to be the case with other SOTA approaches, so likely not the authors' fault), and the improvements over SOTA are small and sometimes mixed depending on which metric one looks at.

**Strengths:**

- significant problem domain where much improvement remains possible

- very well explained technical approach

- great to see asymptotic complexity analysis of the various stages of the approach

- convincing evaluation on the nuScenes dataset

- experiments on increasing temporal information, SSM vs deformable attention, feature normalization, scaling, and feature traversal order are interesting and convincing.

**Weaknesses:**

- table 1, the proposed model does not outperform SOTA in all metrics. This table needs to be described better. If NDS is the main metric, then what about Focal-DETR? Should explain more clearly which table rows are compared in the main text lines 395-396

- should include a figure with reconstruction samples, for qualitative evaluation by the reader.

- lines 221-222: nonsense sentence

**Questions:**

- only tested on nuScenes (like some of the other SOTA models, including DETR3D and BEVformerV2). Is there another suitable dataset that this could be tested on to better demonstrate generality?

---

> ### Comment · Area_Chair_RHLe · 2024-12-01
>
> Dear authors,
>
> Are you sure to leave this reviwer's comments alone?
>
> Thanks,
> AC

---

> > ### Author Response · Authors · 2024-12-01
> > **Response to Area Chair**
> >
> > Thank you so much for your reminder! We are pleased to share that we have responded to all the reviewers' comments.

---

> ### Author Response · Authors · 2024-12-01
> **Response to reviewer yMTq**
>
> Thank you for your thoughtful review, valuable feedback, and the time taken to provide constructive feedback to strengthen our work further! We deeply appreciate your recognition of our contributions, particularly in building an efficient camera-only BEV and conducting convincing ablation experiments to demonstrate the effectiveness of MamBEV. Below, we address your concerns and questions in detail.
>
> > **"table 1, the proposed model does not outperform SOTA in all metrics. This table needs to be described better..."**
>
> Sorry for the confusion!  We have revised our paper to provide an improved description of our main results section, as well as to address the other concerns you mentioned.
>
> In this paper, we trained only the tiny and small versions of our model, maintaining scaling consistent with BEVFormer’s scaling strategy. Notably, our small version, MamBEV-Small, outperforms BEVFormer-Base. Regarding the NDS comparison, we acknowledge there may have been a misunderstanding. Our MamBEV-Small (using 1 frame) was compared to Focal-PETR’s performance. However, it’s important to note that our mAP (mean average precision) achieves the best performance in the 1-frame scenario. Since mAP is weighted five times higher than the other five metrics (ATE, ASE, AOE, AVE, and AAE) in the NDS calculation, this underscores the strength of MamBEV-Small in this specific setting.
>
> In addition to demonstrating the effectiveness of our model, we also highlight its efficiency. We test the memory and compute estimated FLOPs for model configurations which use our XQSSM, standard dot product attention, or deformable attention. XQSSM and deformable attention scale linearly in memory and computational complexity with respect to the size of the inputs $V$ and $Q$, though the coefficient factor of deformable attention is smaller.
>
> | **Cross Attention Module** | **BEV Scale (Q)** | **Image Size (V)** | **Params (K)** | **GFLOPs** | **Memory (GB)** |
> |-----------------------------|-------------------|--------------------|----------------|-------------|-----------------|
> |                             | 50x50            | 800x450           | 239            | 3.7         | 1.7             |
> | **XQSSM**                  | 100x100          | 1280x720          | 239            | 14.0        | 3.5             |
> |                             | 200x200          | 1600x900          | 239            | 51.0        | 6.5             |
> |-----------------------------|-------------------|--------------------|----------------|-------------|-----------------|
> |                             | 50x50            | 800x450           | 156            | 3.3         | 1.7             |
> | **Deformable**             | 100x100          | 1280x720          | 156            | 12.8        | 3.2             |
> |                             | 200x200          | 1600x900          | 156            | 49.5        | 4.9             |
> |-----------------------------|-------------------|--------------------|----------------|-------------|-----------------|
> |                             | 50x50            | 800x450           | 263            | 23.9        | 2.2             |
> | **Dot Product**            | 100x100          | 1280x720          | 263            | 228.8       | 9.7             |
> |                             | 200x200          | 1600x900          | 263            | 1,432.5     | >24             |
>
> ---
>
> We believe this design significantly improves the computational and memory efficiency of the XQSSM module, aligning with the broader goals of efficient 3D spatial awareness in Mamba-2. Thank you again for the opportunity to refine our explanations!
>
>
> > **"Should include a figure with reconstruction samples, for qualitative evaluation by the reader."**
>
> > **"Lines 221-222: nonsense sentence"**
>
> Thank you so much for the suggestion! .We have included five detection visualiztion results in our revised paper, particularly in Figure 5, 6, 7, 8, 9. And we have revised the nonsense sentence.

---

### Official Review · Reviewer_knmh · 2024-11-04

**Soundness:** 2
**Presentation:** 3
**Contribution:** 2
**Rating:** 6
**Confidence:** 4

**Summary:**

The paper presents MamBEV, a novel 3D object detection framework designed to improve BEV-based perception systems, particularly for autonomous driving. By introducing state-spatial separation and multi-axis multiview projection, MamBEV addresses the challenges in handling multi-camera BEV representation. The proposed method achieves promising results in complex driving scenarios, showcasing improvements in detection accuracy and computational efficiency.

**Strengths:**

1. Lower computational cost.
2. Good performance, comparable to VideoBEV.

**Weaknesses:**

1.The paper could have provided more ablation studies to dissect the contributions of individual components of MamBEV.
2. Limited discussion on potential limitations of the framework, such as scalability or adaptability to other domains beyond autonomous driving.
3. limit novelty. The BEVFormer architecture introduces Mamba.

**Questions:**

1. An important reason for using Mamba is to achieve lower computational cost; however, the paper only provides theoretical calculations without experimental validation, which is a crucial aspect.
2. In the bottom experiments of Table 1, why do BEVFormerV1-Small and BEVFormerV1-Base use different frames?

---

> ### Author Response · Authors · 2024-11-30
> **Response to reviewer knmh**
>
> Thank you for your insightful review and feedback. We are delighted that you found our encoder architecture and novel spatial cross mamba block. We are also pleased that our emphasis on effectiveness while maintaining the efficiency of existing models resonated with you. Below, we address your concerns and comments in detail:
>
> > **"The paper could have provided more ablation studies to dissect the contributions of individual components of MamBEV."**
>
> Thank you for your thoughtful suggestion! We have added new ablation experiments to dissect the effectiveness of each individual component of MamBEV. The main new inclusions test the effect of the self attention component, and the insertion position of the queries.
>
> 1. ### Performance Comparison of Self-Attention Styles
>
> We conduct experiments to show the effect of various self-attention styles for the BEV query grid. The results are summarized in the table above. Along with a slight increase in parameters, the performance of Hydra (bi-directional Mamba-2) outperforms both standard self-attention and deformable self-attention. This is evidenced by the improved **NDS** and **mAP** scores, which validate the effectiveness of our BEV Self-Attention design.
>
>
> | **Method**        | **NDS $\uparrow$** | **mAP $\uparrow$** | **mATE $\downarrow$** | **mASE $\downarrow$** | **mAOE $\downarrow$** | **mAVE $\downarrow$** | **mAAE $\downarrow$** | **Params $\downarrow$** |
> |--------------------|--------------------|--------------------|-----------------------|-----------------------|-----------------------|-----------------------|-----------------------|-------------------------|
> | **Self-Attention** | 0.2991            | 0.2196            | 0.9054               | **0.3091**            | 0.7498               | 0.9148               | 0.2274               | **37.5M**              |
> | **Deformable**     | 0.2992            | 0.2246            | 0.9216               | 0.3114               | 0.7437               | **0.9092**           | 0.2447               | 37.5M                  |
> | **Hydra $\alpha=2$** | 0.3061           | 0.2282            | 0.8962               | 0.3131               | 0.7167               | 0.9184               | 0.2358               | 38.1M                  |
> | **Hydra $\alpha=4$** | **0.3128**       | **0.2362**        | **0.8882**           | 0.3098               | **0.7158**           | 0.9114               | **0.2269**           | 38.9M                  |
>
> ---
>
> 2. ### The effectiveness of Spatial Cross Mamba
>
> This table compares encoder layer submodule formulations. Minimal differences are observed between the methods. **Mixed attention** involves a single layer of spatial cross Mamba followed by a single layer of deformable cross attention. While **Spatial Cross Mamba** achieves the highest **NDS** and **mAP**, the **Mixed** method shows improvements in **mATE**, **mASE**, and **mAAE**, highlighting the trade-offs between these approaches. Spatial cross mamba can serve as a replacement for deformable attention in BEV encoding as there is minimal difference in performance. In our experiments, we found that using mixed spatial and deformable attention was helpful when training a larger network that learns representations over a larger number of frames.
>
>
> | **Method**               | **NDS $\uparrow$** | **mAP $\uparrow$** | **mATE $\downarrow$** | **mASE $\downarrow$** | **mAOE $\downarrow$** | **mAVE $\downarrow$** | **mAAE $\downarrow$** |
> |--------------------------|--------------------|--------------------|-----------------------|-----------------------|-----------------------|-----------------------|-----------------------|
> | **Deformable Attention** | 0.3140            | 0.2384            | 0.8891               | 0.3097               | 0.7166               | **0.9107**           | 0.2255               |
> | **Spatial Cross Mamba**  | **0.3141**        | **0.2386**        | 0.8882               | 0.3098               | 0.7158               | 0.9114               | 0.2269               |
> | **Mixed**                | 0.3128            | 0.2362            | **0.8799**           | **0.3086**           | **0.7044**           | 0.9349               | **0.2246**           |
>
> ---
> **<CONTINUED>**

---

> ### Author Response · Authors · 2024-11-30
> **Response to reviewer knmh**
>
> 3. ### Performance Comparison of Query Insertion Methods
>
> This table presents a performance comparison of our model using different query insertion methods:
> - **Append** and **Prepend** represent naive methods of appending/prepending the queries after the corresponding image feature maps.
> - **Project** is our proposed **BEV Position Aware Merge** method.
>
> The **Project** method consistently achieves the best performance across key metrics such as **NDS**, **mAP**, **mATE**, **mASE**, and **mAOE**, demonstrating the effectiveness of our approach. However, **Append** achieves the lowest **mAAE**, and **Prepend** results in the lowest **mAVE**.
>
> | **Method**  | **NDS $\uparrow$** | **mAP $\uparrow$** | **mATE $\downarrow$** | **mASE $\downarrow$** | **mAOE $\downarrow$** | **mAVE $\downarrow$** | **mAAE $\downarrow$** |
> |-------------|--------------------|--------------------|-----------------------|-----------------------|-----------------------|-----------------------|-----------------------|
> | **Append**  | 0.3051            | 0.2314            | 0.8897               | 0.3103               | 0.7560               | 0.9229               | **0.2221**           |
> | **Prepend** | 0.3073            | 0.2300            | 0.8917               | 0.3106               | 0.7695               | **0.8887**           | 0.2265               |
> | **Project** | **0.3128**        | **0.2362**        | **0.8882**           | **0.3098**           | **0.7158**           | 0.9114               | 0.2269               |
>
> ---
> > **"Limited discussion on potential limitations of the framework, such as scalability or adaptability to other domains beyond autonomous driving."**
>
> Thank you for your thoughtful suggestion!  The cross attention mechanism is expected to apply in other domains, but its effectiveness and efficiency are determined by how close the queries are to the relevant information in the values. As the BEV aware merge mechanism we propose is specifically for BEV applications, the adaptation we made is only well suited for spatial cross attention tasks. For other tasks, it may be necessary to add a module similar to that of Deformable Attention which predicts which location in the target sequence the query should be inserted, however such a module was not explored in this work.
>
> For scalability, we test the memory and compute estimated FLOPs for model configurations which use our XQSSM, standard dot product attention, or deformable attention. XQSSM and deformable attention scale linearly in memory and computational complexity with respect to the size of the inputs $V$ and $Q$, though the coefficient factor of deformable attention is smaller. The results are shown in below table:
>
> | **Cross Attention Module** | **BEV Scale (Q)** | **Image Size (V)** | **Params (K)** | **GFLOPs** | **Memory (GB)** |
> |-----------------------------|-------------------|--------------------|----------------|-------------|-----------------|
> |                             | 50x50            | 800x450           | 239            | 3.7         | 1.7             |
> | **XQSSM**                  | 100x100          | 1280x720          | 239            | 14.0        | 3.5             |
> |                             | 200x200          | 1600x900          | 239            | 51.0        | 6.5             |
> |-----------------------------|-------------------|--------------------|----------------|-------------|-----------------|
> |                             | 50x50            | 800x450           | 156            | 3.3         | 1.7             |
> | **Deformable**             | 100x100          | 1280x720          | 156            | 12.8        | 3.2             |
> |                             | 200x200          | 1600x900          | 156            | 49.5        | 4.9             |
> |-----------------------------|-------------------|--------------------|----------------|-------------|-----------------|
> |                             | 50x50            | 800x450           | 263            | 23.9        | 2.2             |
> | **Dot Product**            | 100x100          | 1280x720          | 263            | 228.8       | 9.7             |
> |                             | 200x200          | 1600x900          | 263            | 1,432.5     | >24             |
>
> ---

---

> ### Author Response · Authors · 2024-11-30
> **Response to reviewer knmh**
>
> > **"Limit novelty. The BEVFormer architecture introduces Mamba."**
>
> Apologies for the lack of clarity! To the best of our knowledge, BEVFormer architecture did not introduce Mamba.  To clarify, our novelty includes:
> -  We propose an SSM-based architecture that can exceed the performance of existing Transformer-based architectures.
> -  We propose a novel approach, Spatial Cross Mamba, analogous to standard cross-attention, where a set mapping mechanism enables the association and fusion of two different modalities. In our case, BEV query representations are matched with corresponding image features to facilitate direct integration of information from both modalities.
> - We design a novel SSM module that can fuse two distinct image representations effectively.
> - A thorough set of ablation studies is provided to showcase model scaling and other properties.

---

> > ### Author Response · Authors · 2024-12-02
> > **Response to reviewer knmh**
> >
> > As the discussion stage is coming to a close, we hope our response has effectively addressed your questions and concerns. If so, we would sincerely appreciate it if you could consider raising the score. Thanks again for your very constructive and insightful feedback!

---

> ### Author Response · Authors · 2024-12-03
> **Follow up on our rebuttal**
>
> We hope our response addresses your concerns. If so, we’d greatly appreciate it if you could consider raising the score. Thank you for your valuable feedback!

---

### Official Review · Reviewer_wMH8 · 2024-11-10

**Soundness:** 2
**Presentation:** 2
**Contribution:** 2
**Rating:** 5
**Confidence:** 4

**Summary:**

The main content of the paper is focused on enabling state space models (SSMs) to learn birds-eye-view (BEV) representations, specifically in the context of 3D representation learning tasks.

The authors propose a novel method called MAMBEV, which incorporates SSMs and linear attention to address the challenges of capturing temporal and spatial relationships in multiview video and fusing distinct visual representations.

They conduct experiments using the nuScenes dataset and achieve results comparable to the state of the art.

**Strengths:**

1. This paper presents a method for applying Mamba to BEV detection.

**Weaknesses:**

1. **Writing Issues:**
The paper has several writing issues:
* Many sentences are lengthy, which hinders comprehension. For example, lines 188, 191, and 201.
* Certain sections lack citations, such as on lines 206 and 207.
* Some areas seem to be missing punctuation, possibly in line 225.

2. **Experimental Results:**
The overall improvement in the experimental results is quite limited, and the value of applying SSM to BEV detection is not clear.

3. **Comparison with Related Work:**
Although the authors argue that it is unnecessary to compare with some long-sequence works, I believe it would be best to provide such comparisons, especially given that the authors conducted multi-frame ablation studies. Including comparisons with related work, such as VideoBEV, would strengthen the paper’s claims.

4. **Efficiency Metrics:**
The paper mentions reducing computational load and improving efficiency. The proposed method should provide relevant metrics related to training and testing, such as training time and model latency during testing. While the authors provide some data, these metrics do not demonstrate any significant advantage over more established methods like deformable attention. This lack of practical efficiency undermines the paper’s claims.

5. **Test Dataset Results:**
The authors have not provided results on the test dataset. Their explanation for this omission is unconvincing and raises questions about the generalizability of the proposed method.

6. **Summary:**
Overall, this paper discusses the application of Mamba in BEV detection. However, from the perspective of the detection task, the introduction of Mamba does not demonstrate any practical significance. Combined with the above points, I believe the paper requires more comprehensive experiments to substantiate its contributions.

**Questions:**

Please refer to section Weaknesses.

---

> ### Author Response · Authors · 2024-11-30
> **Response to reviewer  wMH8**
>
> Dear Reviewer,
>
> Thank you for your thoughtful review, valuable feedback, and the time taken to provide constructive feedback to strengthen our work further! We are grateful that you identified writing issues in paper, the efficiency of our model, and the comparison with VideoBEV. We will address your concerns and questions below.
>
> > **"The paper has several writing issues: ..."**
>
> Thank you for your careful reading of our paper. We have fixed the writing issues you highlighted in the revised paper.
>
> > **"The overall improvement in the experimental results is quite limited, and the value of applying SSM to BEV detection is not clear."**
>
> Thank you for your question. We propose a novel approach, Spatial Cross Mamba, analogous to standard cross-attention, where a set mapping mechanism enables the association and fusion of two different modalities. In our case, BEV query representations are matched with corresponding image features to facilitate direct integration of information from both modalities. And we show the effectiveness and efficiency of our proposed method that is linearly in memory and computational complexity.
>
> > **"The experimental comparison is not entirely fair, as the pretrained model used in this study differs from other pretrained models, which may affect the results."**
>
> Thank you for your question. Pretrained models for our primary comparisons BEVFormerV1 and BEVFormerV2 make use of the exact same backbone weights. For other models, the comparison is made on the basis of using the same size backbone -- ResNet50 or ResNet101 depends on the comparison with MamBEV-Tiny or MamBEV-Small, the same (or similar) image resolution, and comparable numbers of frames. We minimized potential variations caused by differences in pretrained models, ensuring that the comparison focuses solely on the performance of the methods themselves.
>
> > **"There is a lack of comparison with more results, such as VideoBEV, which is also a long-sequence application but was not included in the comparison."**
>
> Thank you for your question. Our work does not aim to expand the temporal window of BEV perception models rather on adapting the SSM model family to cross attention tasks. The long sequences which we refer to in our work are the flattened image feature maps and BEV query grids being processed with Mamba. We did perform an ablation on the size of the temporal window, which showed our model was able to adapt information from previous frames through the use of a temporal encoder. However, this component was not a new contribution of our work.  In contrast, VideoBEV’s focus was primarily on providing a novel temporal encoder which allowed for longer temporal windows with minimal computational overhead when compared with modules like that of BEVformerV2. The goal of our main comparison table is to demonstrate that models of similar architectures such as BEVFormer have similar or worse performance to our novel encoder layer.
> These reasons may not completely disqualify VideoBEV as a baseline, however we were not able to construct a fair comparison (same backbone, temporal window, and resolution) as they did not release their code.

---

> ### Author Response · Authors · 2024-11-30
> **Response to reviewer wMH8**
>
> > **"The paper mentions reducing computational load and improving efficiency. Therefore, the proposed method should provide relevant metrics related to training and testing, such as training time and model latency during testing."**
>
> Thank you for your question.
>
> Mamba-2 offers **dual methods of computation** for selective state space models:
> 1. **Matrix Mixer (SSD)**: Implements structured matrix mixers for fast, parallel computation using batch matrix multiplication.
> 2. **Associative Scan (SSM)**: Allows efficient sequential computation through shortened associative scans.
>
> The reframing of SSMs as structured matrix mixers, denoted as \($\mathcal{M}$), provides enhanced computational efficiency through:
> - **Parallelism**: Enabled by batch matrix multiplication.
> - **Flexibility**: Simplified operations with structured matrix formulations.
>
> In practice, we utilize **SSD** during both training and inference because it is better optimized for the hardware used. However, the **kernel** was not adapted to reflect the true computational complexity of the **Cross Quasi-Separable State Space Model (XQSSM)** module.
>
> ---
>
> ### Sequential Implementation and Query Discretization
>
> Algorithm 2 (refer to Appendix) outlines a **sequential implementation** leveraging the unique formulation of our module. This includes **query discretization**, where the query input \(Q_{dt}\) has a discretization factor of \(dt = 0\). In this case, **no hidden state update occurs**, as demonstrated below:
>
> $\begin{equation}
> h_t = \exp(dt_t A) h_{t-1} + dt_t B_t x_t,
> \end{equation}$
>
> $\begin{equation}
> h_t = \exp(0) h_{t-1} + 0,
> \end{equation}$
>
> $\begin{equation}
> h_t = h_{t-1}.
> \end{equation}$
>
> When \(dt = 0\), this reduces the number of operations per query from:
>
>
> 2H(N+1) + $\alpha$ D(3N + H + 1)
>
>
> to:
>
>
> $\alpha$ D(N + H + 1).
>
>
> ---
>
> ### Computational Complexity of XQSSM
>
> Outputs from the XQSSM are only required for **query token inputs**, further reducing the **per-image feature complexity** to:
>
>
> 2H(N+1) + 2$\alpha$ DN.
>
>
> The **total computational complexity** of the XQSSM is:
>
>
> 2V (H(N+1) + $\alpha$ DN ) + M $\alpha$ D(N + H + 1),
>
>
> where:
> - \(M\): Number of queries added to the sequence,
> - \(V\): Number of input features,
> - \(H\): Hidden state size,
> - \(N\): Input sequence length,
> - \($\alpha$ D\): Complexity scaling factor.
>
> The **memory complexity** of the sequential form is **constant**, while in parallelized form, it becomes **linear** with respect to the sequence length, as shown in [1].
>
> ---
>
> ### Matrix Mixer Comparison
>
> The resulting matrix mixer, \($\mathcal{M}\$), reduces in dimensionality from:
>
>
> (M+V) $\times$ (M+V) $\times$ 2H
>
>
> to:
>
>
> M $\times$ V $\times$v2H,
>
>
> which is comparable to the matrix mixer for **dot product cross attention** with a shape of:
>
>
> Q $\times$ K $\times$ H.
>
> In the following table, we test the memory and compute estimated FLOPs for model configurations which use our XQSSM, standard dot product attention, or deformable attention. XQSSM and deformable attention scale linearly in memory and computational complexity with respect to the size of the inputs $V$ and $Q$, though the coefficient factor of deformable attention is smaller.
>
> | **Cross Attention Module** | **BEV Scale (Q)** | **Image Size (V)** | **Params (K)** | **GFLOPs** | **Memory (GB)** |
> |-----------------------------|-------------------|--------------------|----------------|-------------|-----------------|
> |                             | 50x50            | 800x450           | 239            | 3.7         | 1.7             |
> | **XQSSM**                  | 100x100          | 1280x720          | 239            | 14.0        | 3.5             |
> |                             | 200x200          | 1600x900          | 239            | 51.0        | 6.5             |
> |-----------------------------|-------------------|--------------------|----------------|-------------|-----------------|
> |                             | 50x50            | 800x450           | 156            | 3.3         | 1.7             |
> | **Deformable**             | 100x100          | 1280x720          | 156            | 12.8        | 3.2             |
> |                             | 200x200          | 1600x900          | 156            | 49.5        | 4.9             |
> |-----------------------------|-------------------|--------------------|----------------|-------------|-----------------|
> |                             | 50x50            | 800x450           | 263            | 23.9        | 2.2             |
> | **Dot Product**            | 100x100          | 1280x720          | 263            | 228.8       | 9.7             |
> |                             | 200x200          | 1600x900          | 263            | 1,432.5     | >24             |
> ---
>
>
> ## **Reference**
>
> [1] Tri Dao and Albert Gu. Transformers are ssms: Generalized models and efficient algorithms through structured state space duality. arXiv preprint arXiv:2405.21060, 2024.
>
> **<CONTINUED>**

---

> ### Author Response · Authors · 2024-11-30
> **Response to reviewer wMH8**
>
> We believe this design significantly improves the computational and memory efficiency through the XQSSM module, aligning with the broader goals of efficient 3D spatial awareness in Mamba-2. Thank you again for the opportunity to refine our explanations!

---

> > ### Comment · Reviewer_wMH8 · 2024-12-01
> > **Response to author's rebuttal**
> >
> > Thank you for your response. Your feedback has addressed some of the concerns. However, there are still a few points I remain uncertain about:
> >
> > 1. Based on the experimental results, the overall performance improvement is indeed quite limited.
> >
> > 2. One thing I overlooked during the review process is that it seems the paper lacks results on the test set?
> >
> > 3. I still don’t fully understand the value of applying SSM to the detection task. In fact, we shouldn’t apply something to a task just for the sake of applying it, but rather analyze the pain points of the task and how the new architecture can address those pain points. From this perspective, I still find the contribution lacking in innovation.
> >
> > 4. It's not that I insist on comparing with methods like VideoBEV, but since the paper includes an ablation study on temporal aspects, it should be compared with related works on temporal processing, unless the paper doesn’t include any multi-frame settings for comparison at all.
> >
> > 5. Instead of providing FLOPs, I would prefer to see actual runtime results, as runtime is often more persuasive than theoretical computations.
> >
> > In summary, I will maintain my rating.

---

> > > ### Author Response · Authors · 2024-12-02
> > > **Response to points 1 & 2**
> > >
> > > 1- We are seeking to adapt SSMs (a modern RNN-like family of models), to applications which have been dominated by Transformers. Our primary contribution and goal is not a minor improvement in performance in BEV detection, but rather a method to show how Mamba can be used to achieve that performance while maintaining its advantages (linear-cost attention with parallel training).
> > > In the table below, we compare a naïve implementation and our specialized implementation for BEV perception. In this case, the input sequence is relatively short (375 image feature vectors) so a hidden state of size 256 < 375 is likely to capture all relevant information in the sequence. Compared to our result using projection, method we see a small 0.3% increase in NDS performance while having a per block increase in  GFLOPs of 461%, a per block increase in parameters of 98%, and a total memory increase of 10%. The stark difference in efficiency shows the benefit of our work to efficient applications of Mamba for cross attention in BEV and offers direction for future applications to other areas.
> > >
> > > |    Method    |   NDS ↑    |   mAP ↑    | Params (K) ↓ (Spatial Cross Mamba) | GFLOPs ↓ (Spatial Cross Mamba) | Memory (MB) ↓ (Total) |
> > > | :----------: | :--------: | :--------: | :--------------------------------: | :----------------------------: | :-------------------: |
> > > | Append h=32  |   0.3051   |   0.2314   |                240                 |              2.6               |          438          |
> > > | Append h=256 | **0.3138** | **0.2371** |                476                 |              14.6              |          480          |
> > > | Project h=32 |   0.3128   |   0.2362   |                240                 |              2.6               |          438          |
> > >
> > > 2-
> > > Our objective in providing other works is to contextualize ours in a way that is fair and unbiased. We aim to do that by using small existing models as a basis for our overall pipeline configuration. The test set does not help us to make useful comparisons for 2 main reasons: restricted number of submissions, and limited existing submissions for models of a comparable size. Submissions to the test set are restricted to a maximum of three per year, which limits our ability to evaluate smaller-scale or intermediate models in that setting, particularly for models which we did not create. The few test set evaluations are typically used by other authors for the largest and best-performing model configurations. Given that our current largest model is small in scale, such comparisons would be less useful in contextualizing our model's performance, so it was forgone.
> > >
> > > The other primary reason to use the test set is check for leakage from the validation set, which we avoided by limiting our hyperparameter search. For our small model, we trained less than 10 versions in total based on our ablation results for MamBEV-Tiny, and of these two were discarded due to bugs during training. Additionally, only a single configuration was used to train MamBEV-Small Pure. For this reason, we do not believe that any leakage occurred from the validation set. In summary, the test set did not seem likely to provide new or compelling evidence of the effectiveness of the model, so it was not used.

---

> > > ### Author Response · Authors · 2024-12-03
> > > **Response to point 5 & summary**
> > >
> > > 5 - In our assessment, the reported runtime numbers may not accurately reflect the potential of our method, as the implementation we used has not been optimized to the same extent as existing methods. Notably, the authors of Mamba and Mamba-2 have demonstrated that the inner SSM can be computed efficiently and have provided optimized implementations of this block in their works. However, we did not modify this block and instead repurposed the existing Mamba implementation as explained in section 3.1.3. This highlights the possibility for further refinement and performance improvements in our approach.
> > > When compared to attention-based methods, Mamba exhibits significantly better scaling, achieving linear complexity. While the full theoretical efficiency outlined in our work has not yet been fully realized, the estimated FLOPs and inference memory metrics serve as the most accurate representations of our method's performance to date. To underscore our runtime performance gains, we provide a comparative analysis with full attention-based methods in the table below.
> > > | Model               | FPS | Memory (GB) |
> > > | ------------------- | --- | ----------- |
> > > | Transformer         | 3.7 | 9.8         |
> > > | Spatial Cross Mamba | 4.7 | 2.2         |
> > > Both models above use 3 layers (cross attention, self attention, ffn), act only on the smallest feature map (23x40), use a 100x100 BEV grid, with a R101 backbone. The FPS is the average number of samples per second processed by the model in evaluation mode on an RTX 4090 GPU. All cameras views are collapsed into a single sequence.
> > > - - -
> > > To summarize our rebuttal, we view this work from the perspective of expanding Mamba more so than directly advancing the SoTA for BEV perception. Our work is novel in that it introduces many to one cross Mamba that is efficient for BEV applications and offers a basis for applications in other areas to do the same. To better clarify these points, we will update our work to better reflect your critiques. We thank you for your thoughtful consideration of our work and timely responses.

---

> ### Author Response · Authors · 2024-12-02
> **Response to points 3 & 4**
>
> 3- BEV is a setting where the input sequences are long (HD images) and computational efficiency is important (edge computing). SSMs are a good candidate architecture for this task as long sequence recall and efficient computing are their most lauded qualities. Our work is focused on expanding SSMs capabilities to cross attention with specific adaptations for BEV. The results in the 3D detection task serve primarily as proof of SSMs effectiveness in cross attention and lay a groundwork for further optimizations and applications in other tasks.
>
> 4- We reiterate that our works focus is not on the temporal aspect of BEV video perception. The ablation of the temporal window size demonstrates that the model architecture scales with a larger window and justifies our separation of 1 frame and >1 frame configurations in Table 1. We do compare with works which are most similar in number of frames and image size in our main results, Table 1, which we provide in part below. We show that when evaluated against models with a similar number of temporal frames we found comparable or better performance for all true positive metrics.
>
>
> | Model              | Backbone  | Frames | NDS   | mAP   | mATE  | mASE  | mAOE  | mAVE  | MAAE  |
> | ------------------ | --------- | ------ | ----- | ----- | ----- | ----- | ----- | ----- | ----- |
> | PolarDETR-T        | ResNet101 | 2      | 0.488 | 0.383 | 0.707 | 0.269 | 0.344 | 0.518 | 0.196 |
> | BEVFormerV1-Small  | ResNet101 | 3      | 0.479 | 0.370 | 0.721 | 0.279 | 0.407 | 0.436 | 0.220 |
> | BEVFormerV1-Base   | ResNet101 | 4      | 0.517 | 0.416 | 0.673 | 0.274 | 0.372 | 0.394 | 0.198 |
> | MamBEV-Small-Pure† | ResNet101 | 4      | 0.506 | 0.412 | 0.676 | 0.281 | 0.400 | 0.470 | 0.187 |
> | MamBEV-Small       | ResNet101 | 3      | 0.523 | 0.415 | 0.656 | 0.281 | 0.379 | 0.340 | 0.192 |
> | MamBEV-Small       | ResNet101 | 4      | 0.525 | 0.423 | 0.662 | 0.280 | 0.386 | 0.354 | 0.183 |

---

> ### Author Response · Authors · 2024-12-03
> **Follow up on our rebuttal**
>
> As the rebuttal stage is coming to a close, we hope our response has effectively addressed your questions and concerns. If so, we would sincerely appreciate it if you could consider raising the score. Thank you for your valuable feedback!

---

> > ### Author Response · Authors · 2024-12-04
> > **Follow up on our rebuttal**
> >
> > As the rebuttal stage is coming to a close, we hope our response has effectively addressed your questions and concerns. If so, we would sincerely appreciate it if you could consider raising the score. Thank you for your valuable feedback!

---

### Author Response · Authors · 2024-12-03
**Global Response**

Dear Reviewers and Meta-Reviewers,


We are grateful for the time you have spent providing us with constructive feedback and thoughtful comments. Your feedback leads to an improved manuscript – thank you! We are pleased to introduce the Spatial Cross Mamba in our MamBEV model, extending the Mamba framework from a self-attention mechanism to a cross-attention mechanism with linear computational complexity. We will also add anything we missed if this paper is accepted.  After the edits and discussions, we are happy to see two reviewers raise their scores.
We summarize the primary strengths as noted by reviewers in their reviews and discussions:

 - A novel method to enable mamba (an SSM) to work for the setting of BEV representation learning.
 - Low computational cost relative to dot-product attention.
 - Good/comparable performance relative to existing methods.


We also addressed the following weakness which are integrated into the manuscript:

- Experiments and results regarding computational cost
- Paper clarity issues (writing, figures)
- Additional ablation studies


Thank you all for your time and valuable comments. We welcome any follow-up discussions and are committed to further improving the paper based on your feedback.

---

### Meta-Review · Area_Chair_RHLe · 2024-12-23

**Metareview:**

This paper proposed Mamba-based framework for BEV perception task. Due to the benefits of Mamba itself, many tasks have been witnessed significant computation efficiency. The results are impressive in terms of NDS on nuScenes. Most concerns raised by reviewers have been well addressed. Among the very key weaknesses, (a) lack of technical details, (b) some figures / descriptions hard to follow, (c) insufficient comparison methods, have been provided after rebuttal. In general, the overall revised manuscript is slightly above the acceptance bar for ICLR. Please incorporate all the necessary revision in the camera-ready version.

**Additional Comments On Reviewer Discussion:**

In particular, one of the five reviewers still remain negative after authors rebuttal. The major concerns were: (a) limitted performance gain; (b) technical clarifications, (c) more ablations. Authors have addressed these concerns fair enough. Although there is no further response from the reviewer, AC read the response nonetheless.

---

### Decision · Program_Chairs · 2025-01-22

Accept (Poster)